Manuscript prepared for Atmos. Chem. Phys.
with version 2014/09/16 7.15 Copernicus papers of the LaTeX class copernicus.cls.
Date: 12 July 2019

# The 2015-2016 Carbon Cycle As Seen from OCO-2 and the Global *In Situ* Network

Sean Crowell[1], David Baker[2], Andrew Schuh[2], Sourish Basu[3,4], Andrew
R. Jacobson[3,4], Frederic Chevallier[5], Junjie Liu[6], Feng Deng[7], Liang Feng[8,9],
Kathryn McKain[3,4], Abhishek Chatterjee[10,11], John B. Miller[4], Britton
B. Stephens[13], Annmarie Eldering[6], David Crisp[6], David Schimel[6], Ray Nassar[12],
Christopher W. O'Dell[2], Tomohiro Oda[10,11], Colm Sweeney[4], Paul I. Palmer[8,9], and
Dylan B. A. Jones[7]

[1]University of Oklahoma, Norman, OK, USA
[2]Cooperative Institute for Research in the Atmosphere, Colorado State University, Fort Collins, CO, USA
[3]Cooperative Institute for Research in Environmental Sciences, University of Colorado Boulder, Boulder, CO, USA
[4]NOAA Earth System Research Laboratory, Boulder, CO, USA
[5]Le Laboratoire des Sciences du Climat et de L'Environnement
[6]Jet Propulsion Laboratory, California Institute of Technology
[7]Department of Physics, University of Toronto
[8]School of GeoSciences, University of Edinburgh
[9]National Centre for Earth Observation, University of Edinburgh
[10]Universities Space Research Association, Columbia, MD, USA
[11]NASA Goddard Space Flight Center, Greenbelt, MD, USA
[12]Climate Research Division, Environment and Climate Change Canada
[13]National Center for Atmospheric Research, Boulder, CO, USA

*Correspondence to:* Sean Crowell (scrowell@ou.edu)

**Abstract.**

The Orbiting Carbon Observatory-2 has been on orbit since 2014, and its global coverage holds the potential to reveal new information about the carbon cycle through the use of top-down atmospheric inversion methods combined with column average $CO_2$ retrievals. We employ a large ensemble of atmospheric inversions utilizing different transport models, data assimilation techniques and prior flux distributions in order to quantify the satellite-informed fluxes from OCO-2 Version 7r land observations and their uncertainties at continental scales. Additionally, we use *in situ* measurements to provide a baseline against which to compare the satellite-constrained results. We find that within ensemble spread, *in situ* observations and satellite retrievals constrain a similar global total carbon sink of 3.7±0.5 PgC, and 1.5±0.6 PgC per year for global land, for the 2015-2016 annual mean. This agreement breaks down on smaller regions, and we discuss the differences between the experiments. Of particular interest is the difference between the different assimilation constraints in the tropics, with the largest differences occurring in tropical Africa, which could be an indication of the global perturbation from the 2015-2016 El Niño. Evaluation of posterior concentrations using TCCON and aircraft observations gives some limited insight into the quality of the different assimilation

constraints, but the lack of such data in the tropics inhibits our ability to make strong conclusions there.

## 1 Introduction

Understanding the global carbon cycle and how it responds to human and natural forcing is a first order requirement for predicting the future trajectory of Earth's climate (Friedlingstein et al., 2013). Our current understanding is embodied in models of the oceans and land biosphere, which characterize processes such as photosynthesis, respiration, nutrient uptake and transport, fire, and chemical cycling, as well as fossil fuel inventories. Measurements of $CO_2$ dry air mole fraction in the atmosphere serve as an integral constraint on the sum of these in the form of a net flux of $CO_2$ to and from the atmosphere at the surface.

Many studies have used atmospheric transport models in conjunction with *in situ* $CO_2$ observations to infer surface fluxes of $CO_2$ (Gurney et al., 2002; Rödenbeck et al., 2003; Peylin et al., 2005; Baker et al., 2006a; Peters et al., 2007; Chevallier et al., 2010a; Schuh et al., 2010; Feng et al., 2011; Basu et al., 2013; Deng et al., 2014; Lauvaux et al., 2016; Feng et al., 2017) at various spatiotemporal scales. And extremely comprehensive review of these studies can be found in Ciais et al. (2010). All of these studies note that the uncertainty in these estimates grows quickly as we move downscale in space and time, particularly for regions in the tropics and southern hemisphere. This is partially due to the errors present in coarse global transport models, and partially due to a paucity of observations outside of North America and Europe.

To improve upon the sparse spatial coverage provided by the *in situ* $CO_2$ network, estimates of column-averaged $CO_2$ mole fraction ($X_{CO2}$) have been derived from a variety of satellite-based instruments. $X_{CO2}$ can be retrieved from high spectral resolution measurements of reflected sunlight. The first space-based instruments designed for this application include ENVISAT SCIAMACHY (Buchwitz et al., 2005), Greenhouse gases Observing SATellite (GOSAT) TANSO-FTS (Kuze et al., 2009), and Orbiting Carbon Observatory-2 (OCO-2) spectrometer (Crisp, 2015).

Three and a half years after launch, $X_{CO2}$ retrievals from OCO-2 are maturing as observational constraints on the carbon cycle. At this time, however, there are only a few publications that utilize the OCO-2 retrievals explicitly for top down flux estimation (Liu et al., 2017). In this work, we investigate the constraint on surface fluxes of $CO_2$ provided by OCO-2 using an ensemble of atmospheric transport inversion frameworks. By characterizing the impact of transport model and inversion method on the flux estimates using our model suite, and by performing separate inversions with each OCO-2 retrieval type (land-nadir, land-glint, ocean-glint) and with traditional *in situ* observations; by doing this we hope to deduce what aspects of our estimates are robust. What is the constraint of OCO-2 on the partitioning of the global land flux between the north and tropics/south? Was the tropical land biosphere responsible for the $CO_2$ outgassing seen globally during

the 2015/2016 El Niño? Are we able to use the OCO-2 retrievals to estimate $CO_2$ fluxes robustly at regional scales?

The manuscript is structured as follows. Section 2 discusses previous work with GOSAT and OCO-2 retrievals. Section 3 outlines the protocol used to define the experiments that were per-
formed, including a description of the data assimilated and data that was used to evaluate the results. Section 4 presents optimized flux estimates and uncertainties from global to regional scales, along with evaluation using independent data, and discusses implications for our understanding of the carbon cycle. Section 5 examines the results in a broader context and suggests a few ways forward to reduce the remaining uncertainties. Finally, Section 6 provides a summary and overall conclusions.

## 2  Background

### 2.1  GOSAT

The Thermal And Near-infrared Sensor for carbon Observation (TANSO) aboard GOSAT is a Fourier Transform Spectrometer (FTS) that measures radiances in the near-infrared (NIR), shortwave in-frared (SWIR), and thermal infrared (TIR) bands. The NIR and SWIR bands are used to retrieve
$X_{CO_2}$ at a spatial scale of approximately 100 km$^2$. GOSAT retrievals have been analyzed by a vari-ety of teams using different schemes for retrieving column $CO_2$ from the measured radiances (Takagi et al., 2014).

GOSAT $X_{CO2}$ retrievals have been used in global $CO_2$ flux inversions by a number of groups. Houweling et al. (2015) compared results from a number of modeling frameworks for 2009-2010
and found that the GOSAT retrievals constraint resulted in a strong annual sink of 1.0 PgC in Europe, in agreement with Reuter et al. (2014) and Reuter et al. (2017), which was balanced mainly by outgassing in Northern Africa. Biases in the GOSAT retrievals were determined to be a potential cause of the large European sink obtained (Feng et al., 2016), as Houweling et al. (2015) also found that the simulated north- south gradient was too large relative to independent data from the HIAPER
Pole-to-Pole Observations (HIPPO, Wofsy (2011)) flight campaign. The initial work in Houweling et al. (2015) is currently being expanded to a longer time period by the GOSAT team to assess the constraint provided by GOSAT and the impacts of biases (Takagi et al, *in prep*).

### 2.2  OCO-2

OCO-2 measures radiances in the spectral bands near $0.765\mu$m, $1.61\mu$m, and $2.06\mu$m. These radi-
ances are returned as 8 distinct soundings across a narrow swath no wider than 10 km. Each sounding has a spatial footprint that is less than 1.29 km by 2.25 km projected onto the surface. This fine spatial resolution is expected to increase the number of cloud-free scenes, and thus allow more successful retrievals with lower errors, as clouds are known to be a source of error in retrievals (O'Dell et al., 2018b). Additionally, this high spatial resolution permits the detection of some systematic biases

which can appear as a set of unrealistically-varying $X_{CO_2}$ over so-called "small areas" (O'Dell et al., 2018b). OCO-2 flies in the EOS Afternoon Constellation (A-Train) with a 705 km sun-synchronous orbit and equator crossing time between 1:21 pm and 1:30 pm local time. The A-Train orbit has a 16-day ground track repeat cycle, which allows for complete global $X_{CO2}$ coverage twice per month, with approximately 150 km horizontal offsets between nearby revisiting orbits. Observations are
made in one of three modes: nadir (looking at the sub-satellite point), directed toward the solar glint spot, or in the so-called target mode.

Both OCO-2 and GOSAT have been extensively evaluated against the Total Carbon Column Observing Network (TCCON) (Wunch et al., 2017). These validation activities reveal systematic errors in both data sets that must be removed using empirical corrections (Wunch et al., 2011). Even after
bias correction, Wunch et al. (2017) demonstrated significant residual bias in the OCO-2 Version 7 glint soundings taken over the high southern latitude oceans. The land nadir and land glint observations contain residual bias (Wunch et al., 2017), but the magnitudes and spatial patterns of that bias are difficult to detect at regional scales with the TCCON network alone. Comparisons to *in situ*-constrained models clearly highlight some of these differences, but it is difficult to distinguish
between bias and real signal in regions with sparse data density.

### 2.3  Flux Estimates with Satellite Observations

In addition to Houweling et al. (2015), numerous other studies have demonstrated that inference of fluxes with atmospheric transport inversions, or "top-down" estimates, can be sensitive to modeled transport (Gurney et al., 2002; Baker et al., 2006a; Stephens et al., 2007; Houweling et al., 2010;
Chevallier et al., 2010b; Nassar et al., 2011; Deng et al., 2015; Basu et al., 2018; Schuh et al., 2019). The covariance of errors due to seasonal sampling and transport has been studied in a series of idealized simulation experiments by Basu et al. (2018), who reported that this can be a significant source of error that may not be reflected in the spread for inversions constrained with OCO-2 retrievals. For example, Figure 5 in Basu et al. (2018) shows that for the boreal regions, the efflux due to the onset
of senescence in the fall is overestimated with the OCO-2 retrievals by more than 0.1 PgC per year, but the spread in flux estimates due to transport is insufficient to differentiate between models and source data. Additionally, Schuh et al. (2019) showed that vertical and meridional mixing differences between two widely used transport models, TM5 and GEOS-Chem, lead to large differences in the inferred northern hemisphere meridional gradient, particularly when separated along the storm track
in the Northern Midlatitudes. These findings, as well as those of Peylin et al. (2013) and others, show that inference using a single model is problematic, and an ensemble of models with varying transport, prior fluxes, and data assimilation methodologies gives an estimate of the sensitivity of inferred flux to the assumptions spanned by the ensemble of models.

## 3 Experimental Design

The work reported here emerges from a large model intercomparison project (MIP) organized by the OCO-2 Science Team in order to understand how flux estimates using OCO-2 retrievals and *in situ* measurements depend on 1) transport, 2) data assimilation methodology, 3) prior flux (and its associated uncertainty) and 4) systematic errors in the OCO-2 retrievals. The OCO-2 MIP is composed of modelers using four different transport models with varying configurations, multiple

different data assimilation frameworks, and diverse prior fluxes and uncertainties. This information is summarized in Table 2 and detailed in the supplementary information. We treat the scatter in the posterior fluxes across this ensemble induced by variability across these parameters as a proxy for the uncertainty in optimized fluxes.

In order to control the drivers of ensemble spread, several assumptions for the different modeling

efforts were standardized. The OCO-2 MIP team utilized a standard 10s average $X_{CO_2}$ values for the time period from September 6, 2014 through April 1, 2017, with appropriate model-data mismatch values as described below to avoid spread due to data handling. Peylin et al. (2013) noted a difference in flux estimates due to different assumed fossil fuel emissions, which are not typically optimized in global top down studies. To avoid this, all group members utilized the same fossil emissions, namely

the Open-source Data Inventory for Anthropogenic $CO_2$ monthly fossil fuel emissions (ODIAC2016; Oda and Maksyutov (2011), Oda and Maksyutov (Reference Date: September 23, 2016), Oda et al. (2018)) together with the TIMES diurnal and weekly scaling (Nassar et al., 2013). The OCO-2 MIP results are connected to other modeling studies such as Transcom (Gurney et al., 2002) and RECCAP (Peylin et al., 2013) through another set of inversions that were performed by each group using a

standardized set of *in situ* measurements (described below).

### 3.1 OCO-2 retrievals

This work utilizes the Version 7 retrospective (V7r) OCO-2 retrieval dataset with a few modifications. The V7 dataset was released in late 2015 and was the first retrieval version from the OCO-2 mission with the precision and accuracy in $X_{CO_2}$ required for scientific use. Initial work with these

retrievals indicated a residual bias that was correlated with regions of high albedos in the $2\mu$m band and relatively low albedos in the $O_2$ A-band. An additional correction was added to reduce the effects of this "s31" bias, which is related to the signal to noise ratio in the $O_2$ band vs. the strong $CO_2$ band. The fine-scale detail contained in individual OCO-2 retrievals is not resolvable by global transport models, which provide $CO_2$ values for large grid boxes that are at least 100km in each

dimension, with specific values given in Table 2. Rather than ingesting each OCO-2 retrieval falling inside a model grid cell separately, we compute a single representative retrieval value for a grid cell with appropriate uncertainty and assimilate that single value. The appropriate uncertainty to assign that representative retrieval is a function of the number of soundings it represents, their individual

uncertainties, representativeness of soundings for the grid box, and the correlations between their individual errors. Since different models use grid boxes of different sizes, we grouped individual retrievals into 10-second bins (groundtrack swaths of 67 km in length), and we assume that the uncertainties between different 10s averages are independent. This assumption is in line with the conclusions of Worden et al. (2017). The spatial scale represented by the 10s averages is small enough to provide enough detail for the highest resolution global models included in this study. The OCO-2 10s sounding locations for nadir and glint retrievals over land are shown in the top row of Figure 1. The number of glint and nadir retrievals varies by month, but the total fraction of good quality retrievals is roughly equivalent between the two modes due to the observing strategy after mid-2015, in which nadir and glint observing modes are interleaved on each orbit that passes over significant land mass. For reference, there are 445113 nadir soundings and 550008 glint soundings in the top two panels of Figure 1 for June 2016, while there are 261380 nadir and 268359 glint observations over land in March 2016. Glint retrievals tend to have larger sun-surface-satellite path lengths, and hence are screened out at higher solar zenith angles when in some cases nadir observations may not be.

Each 10s average consists of a single observing geometry (glint or nadir). In line with the conclusions of Wunch et al. (2017), the ocean glint retrievals are not assimilated due to poorly understood biases, particularly in the high southern latitudes. All OCO-2 experiments detailed in the Results and Discussion sections assimilate land glint and land nadir retrievals only.

We de-emphasize soundings that are taken close together in time and space, since their errors are likely to be strongly correlated. In the absence of a good description of spatial error correlations, we 1) averaged the retrievals into 1-second bins along track (6.7 km) and then 2) averaged all 1s spans with good retrievals within the 10s span to get the 10s values for a given observation geometry. The weighting of each individual value within the 1s and 10s spans is done according to the uncertainty in each sounding, so that assimilating the summary value will give the same result as assimilating the individual values separately (assuming they are independent), although we assign an uncertainty to each aggregate value that is higher to reflect the fact that errors in the individual retrievals are highly correlated, and to account for transport errors.

*Computing the 1s averages*:

We first select only those retrievals in the OCO-2 Lite files (from the "lite_test_20170410" build) with "good" retrievals according to the "xco2_quality_flag" variable. The inverse of the reported single sounding L2 posterior error variances is used to construct a weighted average of many of the variables in the Lite files (time, latitude, longitude,surface pressure, prior, retrieved and bias-corrected $X_{CO_2}$, averaging kernel vector, $CO_2$ vertical profile, pressure weighting function, and independent variables used as part of the bias correction procedure to screen and correct the retrievals)

is computed from these selected retrievals across each 1s span as follows:

$$\widehat{X_{CO_2}} = \sum_i X_{CO_2,i} \sigma_i^{-2} / \sum_i \sigma_i^{-2} \tag{1}$$

where $\widehat{X_{CO_2}}$ denotes the 1s average, $X_{CO_2,i}$ are the values from each sounding, and $\sigma_i$ are the uncertainty in $X_{CO_2,i}$ for each shot (from variable *xco2_uncert*). If each shot in the span were independent, $\widehat{X_{CO_2}}$ would have a theoretical uncertainty of:

$$\sigma_{IND} = 1 / \sqrt{\sum_i \sigma_i^{-2}} \tag{2}$$

where the uncertainty of the average drops approximately by $\sqrt{N}$, where N is the number of shots in the average. However, since we believe the $X_{CO_2}$ retrievals in the small area viewed inside one second are actually highly correlated, we instead use an average uncertainty of the N shots to represent the uncertainty of the average:

$$\sigma_{IVE,1s} = 1 / \sqrt{N^{-1} \sum_i \sigma_i^{-2}} \tag{3}$$

Because even this average uncertainty is sometimes too low (since it captures only the random estimation errors in the retrieval and not any systematic errors), we compare it to the standard deviation of all retrieved $X_{CO_2}$ in the 1s interval, denoted by $\sigma_{spread}$, as well as to a minimum uncertainty threshold (for those cases in which there are too few shots to compute a realistic spread), denoted $\sigma_{floor}$, and we then set the uncertainty for $\widehat{X_{CO_2}}$, denoted by $\hat{\sigma}$, to be the maximum of $\sigma_{IVE,1s}$, $\sigma_{spread}$, and $\sigma_{floor}$.

*Computing the 10s averages:* 10s average values are computed across all 1s spans j with valid retrievals again as the IVE:

$$\overline{X_{CO_2}} = \sum_j \widehat{X_{CO_2}}_j \hat{\sigma}_j^{-2} / \sum_j \hat{\sigma}_j^{-2} \tag{4}$$

Again, we compute the average uncertainty as:

$$\overline{\sigma}_{IVE,10s} = 1 / \sqrt{J^{-1} \sum_j \hat{\sigma}_j^{-2}} \tag{5}$$

where J is the number of 1s values in the sum (just those with good data available). An additional uncertainty representing the variability across models at the OCO-2 sounding locations, denoted $\overline{\sigma}_{model}$ is added in quadrature to $\overline{\sigma}_{IVE,10s}$, and this value is treated as the uncertainty for the 10 s average $\overline{X_{CO_2}}$, which is often referred to as the model-data mismatch (MDM) uncertainty. The MDM is effectively a weighting factor for each retrieval, with small values representing retrievals with the greatest expected utility in the assimilation.

### 3.2 *In situ* $CO_2$ measurements

$CO_2$ measurements collected in flasks or by continuous analyzers at surface, tower, and aircraft sites are an important anchor for this exercise because their error characteristics are generally well-known, being directly established via calibration traceable to WMO standards. Additionally, these measurements provide traceability to a long history of flux estimates derived from these data as an atmospheric constraint. The *in situ* measurements used in these simulations come from the GLOB-ALVIEW+ project (Cooperative Global Atmospheric Data Integration Project, 2016), and from a system developed for this project to deliver near-real time (NRT) $CO_2$ measurements (Carbon-tracker Team, 2017), with spatial locations depicted in Figure 1. Both of these efforts are coordinated by collaborators at NOAA Earth Systems Research Laboratory (ESRL). Each August, the GLOB-ALVIEW+ project publishes a collection of $CO_2$ measurements from academic and institutional data providers covering the previous calendar year. Measurements for this study were compiled from the GLOBALVIEW+ 2.1 and 3.1 releases. As of version 3.1, GLOBALVIEW+ contains more than 14 million individual measurements of $CO_2$ in 353 datasets from 46 contributing laboratories, spanning the time range 1957 to 2016.

Several international measurement networks and campaigns are able to provide $CO_2$ observations with little or no delay, and NOAA has collected and published these measurements from many different sites in the "Near Real Time" (NRT) format. Because many laboratories are not configured to deliver measurements in near-real time (NRT), there are many fewer datasets available in the NRT $CO_2$ product. These include provisional flask measurements from NOAA surface and aircraft sites, made available as soon as laboratory analysis is complete but without final quality-control procedures. Some of the final quality-control analyses require a full year's worth of data. In other cases, analysis of multiple species measured from the same sample of air reveals contamination from local sources; this is a more involved process with longer delays. Among the data streams for NRT measurements are those from NOAA observatories and tall tower systems, and tower sites from Environment and Climate Change Canada. These sites run quasi-continuous analyzers with time-averaged observations being available at approximately hourly frequencies. Other data available in the NRT ObsPack include measurements from the ACT-America (https://act-america.larc.nasa.gov/), OR-CAS (Stephens et al., 2018; Stephens, 2017), and ATom (Wofsy et al., 2018) campaigns. Both GLOBALVIEW+ and NRT $CO_2$ measurement compilations may be downloaded in ObsPack format (Masarie et al., 2014) from https://www.esrl.noaa.gov/gmd/ccgg/obspack/data.php. The observations provided for assimilation in this study are taken from GLOBALVIEW+ v2.1 and NRT v3.3.

Available *in situ* $CO_2$ measurements vary widely in their levels of usable information content and the level to which they can be simulated and interpreted by coarse-resolution global models. To express this level of interpretability, each measurement is assigned a model-data mismatch (MDM) value. For convenience, many modelers have used the "adaptive" model-data mismatch scheme used by the CarbonTracker project (CT2016 release; Peters et al. 2007, with updates documented at http:

//carbontracker.noaa.gov). This scheme is unique in that it assigns temporally-varying MDM values
to account for large seasonal variability in the performance of models. Many measurements are
deemed unsuitable for assimilation into models of this class, due to excessive vertical stratification
during stable planetary boundary layer conditions, proximity to large anthropogenic sources, the
influence of complex terrain, and other reasons.

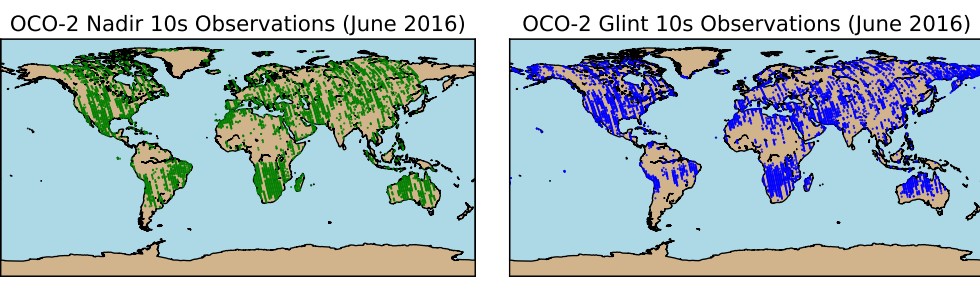

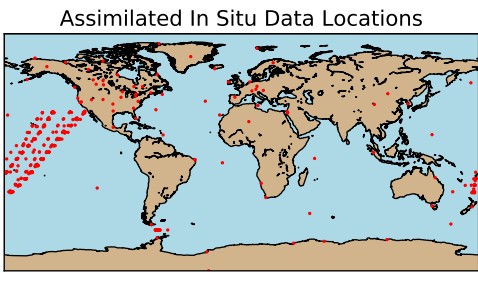

**Figure 1.** Sample locations of different data sources described in the text. Locations of OCO-2 nadir (top left panel) and glint (top right) 10s retrievals for June 2016, *in situ* assimilation data (bottom).

### 3.3 TCCON

The Total Carbon Column Observing Network (TCCON) is a global network of Fourier-transform near-infrared (FTIR) spectrometers that retrieve the column average dry air mole fraction of trace gases such as $CO_2$ and $CH_4$ by analyzing the absorption of incident sunlight. The current version (GGG2014) of $X_{CO_2}$ from TCCON instruments are available at http://tccondata.org/, and a summary of all sites is given in Table 1. For this work, we downloaded all TCCON retrievals available as of July 6, 2017. We filtered the retrievals for outliers and averaged them to create 30 minute average $X_{CO_2}$ values. Details are given in Section C.

### 4 Results and Discussion

Each posterior flux is constrained by a single observation type. Posterior flux estimates are presented for *in situ* observations, with locations shown in Figure 1, and OCO-2 land nadir (LN) and land glint (LG) observations only, due to the obvious bias present in the OCO-2 ocean glint observations as previously mentioned. Ocean nadir data is not provided as a standard data product due to low

**Table 1.** TCCON stations used in this work for evaluation of inverse models.

| TCCON station | Reference |
|---|---|
| Ascension Island | Feist et al. (2014) |
| Bialystok, Poland | Deutscher et al. (2015) |
| Bremen, Germany | Notholt et al. (2014) |
| Caltech, Pasadena, CA, USA | Wennberg et al. (2015) |
| Darwin, Australia | Griffith et al. (2014a) |
| Edwards (Armstrong), CA, USA | Iraci et al. (2016) |
| Eureka, Canada | Strong et al. (2016) |
| Karlsruhe, Germany | Hase et al. (2015) |
| Lamont, OK, USA | Wennberg et al. (2016) |
| Lauder, New Zealand | Sherlock et al. (2014) |
| Manaus, Brazil | Dubey et al. (2014) |
| Orléans, France | Warneke et al. (2014) |
| Park Falls, WI, USA | Wennberg et al. (2014) |
| Réunion Island | De Mazière et al. (2014) |
| Saga, Japan | Kawakami et al. (2014) |
| Sodankylä, Finland | Kivi and Heikkinen (2016) |
| Tsukuba, Japan | Morino et al. (2016) |
| Wollongong, Australia | Griffith et al. (2014b) |

signal to noise ratios in the nadir viewing geometry over the ocean. Unless otherwise stated, prior and posterior fluxes have fossil fuel emissions pre-subtracted, meaning that fluxes over land are the sum of the photosynthesis, respiration, fires, and any effects from land use changes. Details of the
275 different modeling assumptions are summarized in Table 2, and in greater detail in Appendix A.

We present the fluxes at the largest (i.e. global) scales first, and then move to zonal bands, and then finally to regional scales for the regions depicted in Figure 2.

The complete collection of regional flux datasets and imagery, as well as evaluation results, can be found at the OCO-2 MIP portal, found at https://www.esrl.noaa.gov/gmd/ccgg/OCO2/index.php.

### 4.1 Global Flux Estimates

Since $CO_2$ is conserved at the global scale in these simulations, we expect that fluxes at that scale should be well-constrained even with a modest collection of observations. As we see in the left panel of the top row of Figure 3, this is the case. As the right panel shows, all observation types constrain a
285 similar seasonal cycle with comparable peak sinks during the northern hemisphere growing season.

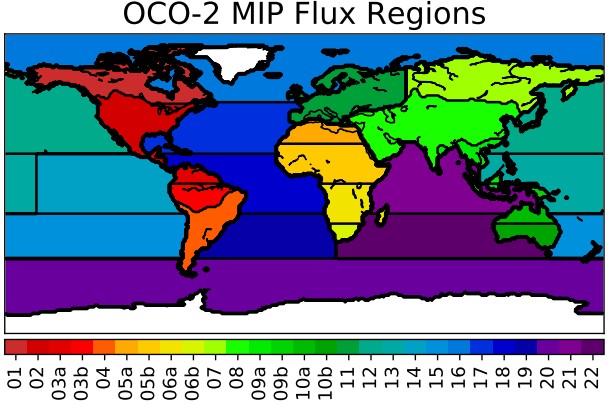

**Figure 2.** Regions on which prior and posterior gridded fluxes are aggregated for comparison.

Interestingly, this peak sink is about 0.75 PgC per month larger than that of the prior emissions, and with a smaller spread. Additionally, all observations lead to a shifted seasonal cycle in which the northern hemisphere growing season begins earlier and ends earlier than assumed in the prior. All data sets produce similar annual mean non-fossil fluxes, -3.5 PgC per year to -4 PgC per year, with a standard deviation of about 0.5 PgC per year across the ensemble. Schuh et al. (2019) showed some dependence of this number upon transport model, implying that further reduction of spread is likely still possible. Additionally, the satellite retrievals suggest a slightly stronger peak growing season sink in 2016 than 2015, though this is not affirmed by the *in situ* measurements and is within the uncertainty as seen in the model spread. The global mean sink for all three results is larger than the results of Peylin et al. (2013) (for 2000-2004) but is consistent with those in Houweling et al. (2015) (for 2009-2010), which agrees with increasing uptake of $CO_2$ by the global land and ocean as deduced by the *in situ*-derived atmospheric growth rate (Ballantyne et al., 2012).

Figure 3 also depicts the global fluxes for land (middle row) and ocean (bottom row) separately. Land fluxes drive the patterns seen in the top row of Figure 3. The summertime drawdown is shifted earlier in the year, and the peak of the drawdown is significantly larger, relative to the prior. Global ocean fluxes are largely unchanged relative to the prior. The shaded regions that pass outside of the prior spread are driven by 3 models that use larger prior uncertainties for ocean fluxes, allowing larger flux increments from atmospheric data, which indicates that the land data could provide some constraint on ocean fluxes were the prior constraint sufficiently weak. This pattern is repeated in the annual ocean fluxes in the left-hand panels.

## 4.2 Zonal Flux Estimates

OCO-2 observes across the sunlit portion of the Earth 14-15 times per day, spanning a large latitudinal range. This fact, combined with the general zonal structure of large scale winds in the atmosphere, suggests that the observations should constrain fluxes in zonal bands. The difference in

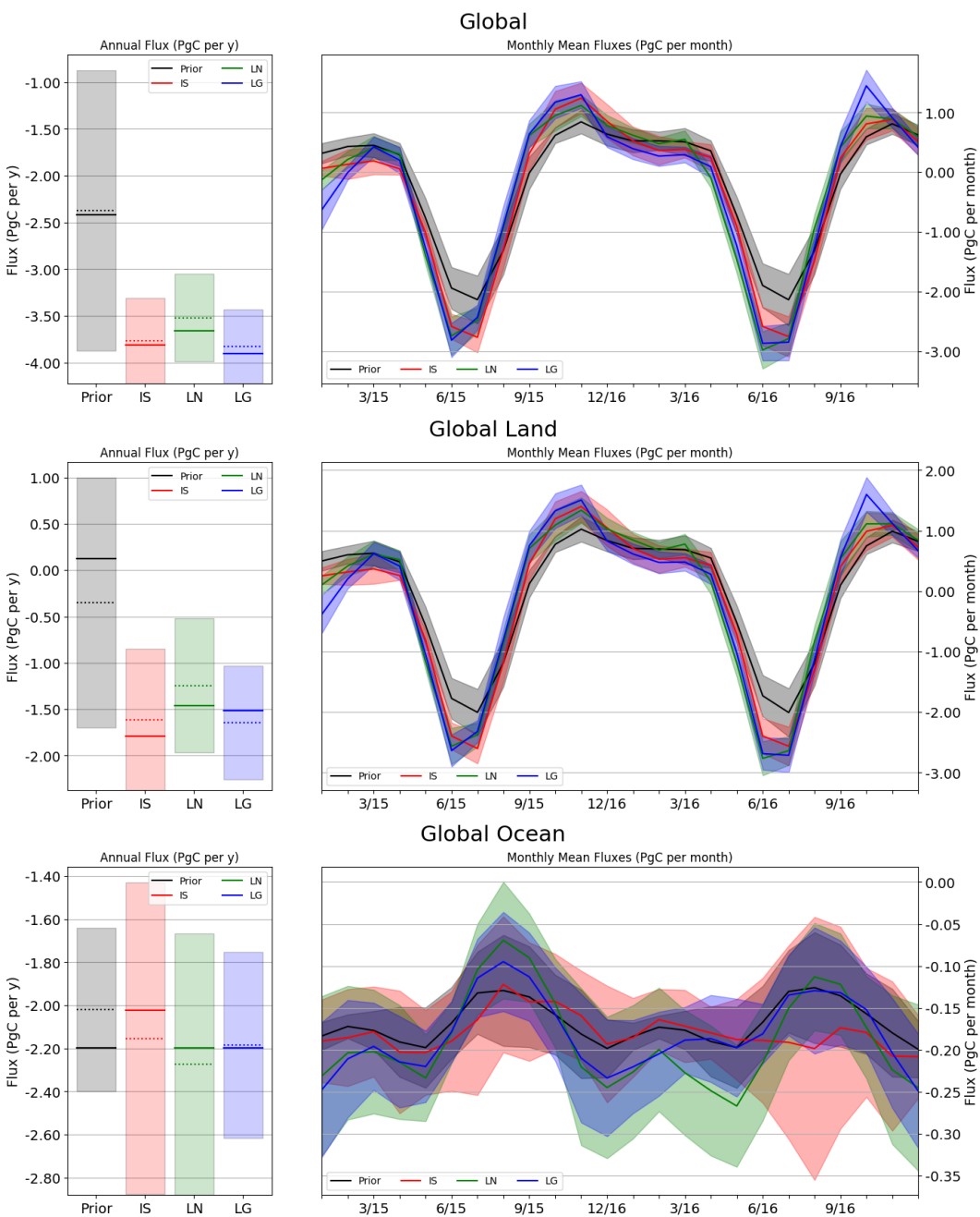

**Figure 3.** Prior (black) and posterior 2015-2016 mean (left) and monthly (right) fluxes constrained by *in situ* (red), OCO-2 land nadir (green), and OCO-2 land glint (blue) observations. (left) The shaded bar represents one standard deviation of the model ensemble about the ensemble mean annual mean flux (dashed line). The solid horizontal line for each bar depicts the median of the ensemble annual mean fluxes. (Right) For each time series, the solid line represents the mean of the OCO-2 MIP ensemble, while the shading represents the ensemble standard deviation.

seasonality in the northern and southern hemispheres, even in the tropics, leads us to examine fluxes split by hemisphere, together with the distinction of tropics and extratropics. Figure 4 shows prior and posterior fluxes at the monthly and annual time scales in the same manner as Figure 3, but split into zonal bands: Northern Extratropics (23N-90N), Northern Tropics (Equator - 23N), Southern Tropics (23S - Equator), and Southern Extratropics (90S - 23S).

The top row of Figure 4 depict the results for the Northern Extratropics. The global seasonality patterns in Figure 3 are reproduced in the Northern Extratropics, with deeper sinks relative to the prior, and a growing season that is shifted earlier in the year. Interestingly, LG fluxes in this region have a weaker annual mean sink (2.6 PgC per year) than the other two experiments (-3 PgC and -3.3 PgC per year for IS and LN, respectively), which is largely driven by enhanced outgassing in the

autumn in 2016. OCO-2 land glint observations are limited to lower latitudes during the NH winter as a result of the longer path lengths than nadir at higher solar zenith angles and high latitudes, and hence there are fewer observations during this time period to constrain the LG results than the other two experiments. Additionally, retrieval biases are expected to grow with sensor and solar zenith angles (O'Dell et al., 2018b), and thus we speculate that this extra outgassing at higher latitudes is

perhaps an artifact of the observations, either due to sampling or retrieval bias.

The Southern Extratropics in the bottom row of Figure 4 are characterized by very little land mass, and hence much less land retrieval data to constrain fluxes. Coupled with the fairly large uncertainty on land fluxes in this region and potential satellite bias at the larger solar zenith angles, we see an unsurprising lack of agreement for each experiment's ensemble. Given the global mini-

mization structure of modern data assimilation systems, it is possible that the fluxes in this region represent a "residual" from matching stronger data constraints in other regions, though this is difficult to test directly. We also note the similar relative differences between the modes, between the Southern Extratropics and the Northern Extratropics, suggesting that biases between modes may drive differences at high latitudes.

The Northern and Southern Tropics are displayed in the middle two rows of Figure 4. OCO-2 observations have potential to significantly improve our understanding of the tropical carbon cycle, given their relatively frequent coverage in a region that is poorly observed by the existing *in situ* network. However, persistent cloudiness during the wet season and biomass burning aerosol in the dry season in the tropics can lead to both fewer observations and residual bias in those that occur in

the vicinity of clouds and aerosols (Merrelli et al., 2015; Massie et al., 2017). Examining Figure 4, we see that the seasonal cycles resulting from the assimilation of OCO-2 data have a larger amplitude seasonal cycle (0.8 PgC per month and 1.5 PgC per month max-min in Tropical Northern Africa and Tropical Southern Africa, respectively) than the inversions in which *in situ* measurements were assimilated (0.6 PgC per month in both regions). The differences in the peak-to-trough fluxes were

determined to be statistically significant for both the Northern and Southern Tropics (not shown). OCO-2 sees a source in 2016 in the Northern Tropics, though the inferred source from the LN

observations is larger than that from LG ($1.5 \pm 0.6$ PgC/y and $0.8 \pm 0.6$ PgC/y for LN and LG, respectively), while the *in situ* measurements place a source of $0.75 \pm 0.6$ PgC in the Southern Tropics. The *in situ* results follow the pattern of the prior at both the monthly and annual time scales, as expected due to the sparse coverage in the tropics, while the amplitude of the satellite data informed fluxes depart significantly from the prior. However, neither the satellite nor the in situ fluxes deviate significantly from the phase of the prior ensemble mean. The results for the annual source in the tropics from LN agree with the findings of Houweling et al. (2015) for GOSAT, being about 1.5 PgC/y for 2009-2010, while the LG fluxes are nearly neutral due to an inferred sink in the Southern Tropics (Houweling et al. (2015) only used nadir data over land from GOSAT). Since 2009-2010 was also an El Niño event (Kim et al., 2011), this suggests that the tropics experienced a similar response to El Niño conditions during those two periods, or that GOSAT and OCO-2 retrievals have similar biases in the Tropics. Importantly, the prior fluxes in our study have a stronger mean tropical source than those in Houweling et al. (2015) (0.7 PgC/y versus 0.3 PgC/y), which may account for the stronger IS source in our study relative to Houweling et al. (2015). In all cases, these conclusions are based primarily on the ensemble mean and spread, and individual models may respond differently, though the comparison of individual models is beyond the scope of this work.

The annual mean flux from the Northern Extratropics and tropics are expected to be strongly anti-correlated with one another across the ensemble, as atmospheric inversions attempt to match the annual growth rate in the global carbon sink. Houweling et al. (2015) found that the surface flask network and GOSAT-constrained meridional gradients were indistinguishable above the ensemble spread, though there is a suggestion of a stronger tropical source. We found that the annual mean flux in the Northern Extratropics and Tropics are also of similar magnitude in the IS, LN and LG experiments when the Northern and Southern Tropics are combined, in agreement with Houweling et al. (2015). The *in situ* measurements used to produce the IS results are different than the data used in Houweling et al. (2015), as are the time periods being studied (2009-2010 vs. 2015-2016). Nonetheless, the flux gradient between the two regions is similar between Houweling et al. (2015) and the results in our study.

### 4.3 Northern Extratropical Region Flux Estimates

The posterior ensembles for the IS, LN and LG experiments exhibit similar seasonality, though different annual sinks, in the Northern Hemisphere extratropical zonal band, and so we examine the fluxes there by continent to determine whether this agreement extends to smaller regions. As is apparent in Figure 5, the different experiments agree over Europe. This contrasts with Houweling et al. (2015), who found that GOSAT retrievals called for a European sink that was much larger than that inferred from *in situ* measurements, though for a different year. North American fluxes show a more complex pattern, with the LN experiment evincing a larger drawdown in 2016 than 2015 that is not present in the other two experiments. Additionally, the annual flux for the LN experiment is

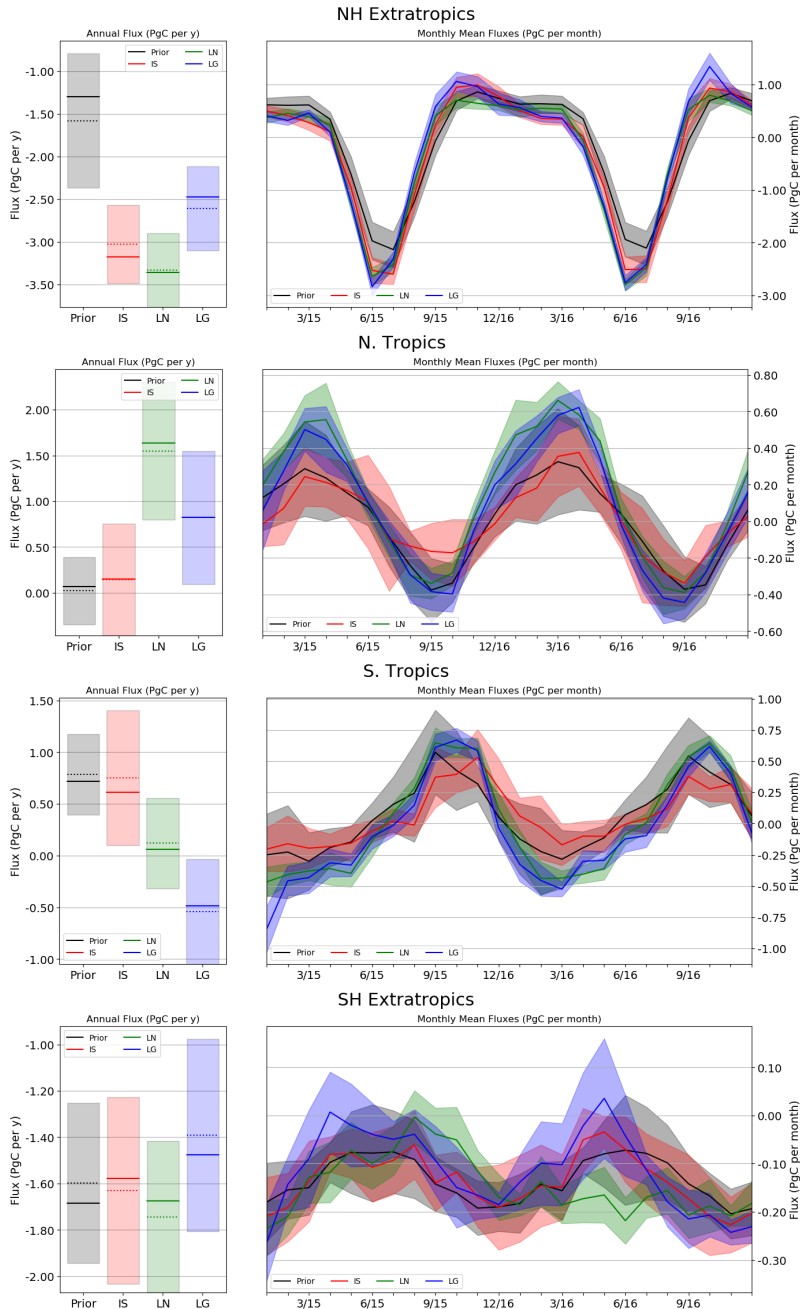

**Figure 4.** As in Figure 3, but for zonal regions described by intervals of latitude: 23°N - 90°N (NH Ext), 0° - 23°N (N. Tropics), 23°S - 0° (S. Tropics), 90°S - 23°S (SH Ext). As in the case of global land, posterior net fluxes in the Northern Extratropics all demonstrate a larger peak drawdown as well as a different seasonality from the prior, with the net drawdown period beginning earlier and ending earlier. The satellite retrievals imply a much more dynamic seasonality than either the prior or the *in situ*-constrained fluxes, particularly in the tropics, where the amplitudes of the seasonal cycles are significantly larger.

less than that from the IS or LG experiments. This is driven by suppressed wintertime efflux for the LN experiment. Interestingly, both sets of OCO-2 retrievals suggest a peak sink that is a month earlier than the *in situ* measurements for both 2015 and 2016. In both Europe and North Asia (i.e. TransCom 7+8), the LG experiment yields a stronger outgassing in the autumn than the other two experiments, which has the same potential explanation as for the Northern Extratropics taken as a whole that was discussed above. Interestingly, both North America (i.e. TransCom 1+2) and North Asia show larger sinks for 2015-2016 than is explicable by the ensemble spread present in Peylin et al. (2013), which could indicate that the sinks in these regions are growing with time, though our experiments encompass only a two year time period that is influenced by the El Niño, and further years of data are required to test this hypothesis.

## 4.4 Tropical Region Flux Estimates

The *in situ* measurements and OCO-2 land retrieval inversions give significantly different results for the two zonal bands focused on the Tropics. In order to gain further insight, we examine fluxes for six smaller regions that compose the signal for these bands to look for meridional information. These regions are subdivisions of the regions from the Transcom 3 project, split at the equator to avoid mixing the seasonality in the Northern and Southern Hemispheres (see Figure 2). The results are displayed in Figure 6 and Figure 7, and demonstrate that the largest differences between the satellite-driven and *in situ*-driven experiments are in Tropical Africa (TransCom 05b+06a), and that the annual fluxes for LN and LG differ most in Tropical Asia (TransCom 09a+09b). Perhaps unsurprisingly, the flux patterns are different north and south of the equator and follow, to a large extent, the phase of the mean prior, which tends towards dry season sources and wet season sinks. In Northern Tropical Africa (TransCom 05b), the difference between the *in situ* and satellite inversions is largely during the drier part of the year (November-March), indicating a much larger source from this region inferred from the OCO-2 retrievals than from *in situ* measurements. In Southern Tropical Africa (TransCom 06a), the OCO-2 experiments indicate a larger amplitude in both dry and wet seasons (which anti-phased with the seasons in Northern Tropical Africa) and some indication of a shift of about a month later in the year for peak carbon efflux. The other four regions are somewhat more difficult to interpret, given the disagreement between models for any of the assimilation constraints. In particular, the different viewing modes of OCO-2 are seeing different things in Tropical South America (TransCom 03b), likely due to residual biases in the observations.

These differences must be interpreted in the context of the density and quality of measurements and the priors. There are more OCO-2 retrievals in this region relative to *in situ* measurements, but there are relatively fewer successful retrievals during the wet season due to the prevalence of clouds. Adjustments to the prior occur mainly during the dry season when there are more satellite measurements, although this is more true for Northern Tropical Africa; significant adjustments from the mean prior in Southern Tropical Africa occur during the wet season as well. Additionally, cloud

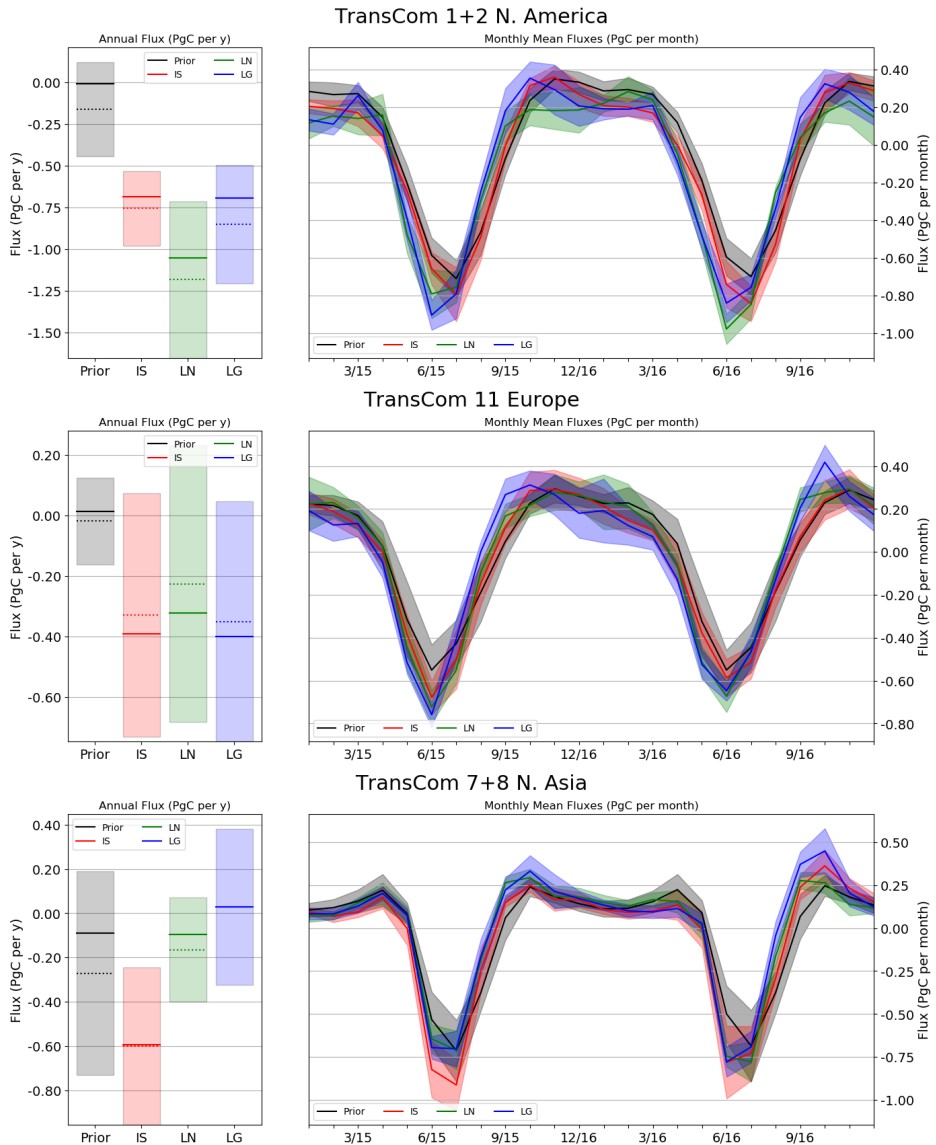

**Figure 5.** As in Figure 3, but for three continental scale land regions in the northern extratropics. The monthly fluxes show broad agreement between the *in situ* and OCO-2 experiments in terms of seasonality and peak drawdown. The fluxes show differences in North America, where the summertime peak sink is larger and wintertime respiration is smaller in the LN experiment results than the other two. The three experiments agree for Europe, which diverges from results in Houweling et al. (2015) in which GOSAT retrievals lead to a much stronger sink in Europe than the *in situ* measurements. In both Europe and Asia, LG experiment results display the enhanced outgassing in the autumn in 2016 present in the northern extratropics seen in Figure 4.

edges could potentially bias retrievals and lead to spurious patterns in the posterior fluxes. This
hypothesis is difficult to reject given the dearth of evaluation data in the tropics.

When Africa as a whole is considered, the total annual $CO_2$ surface emissions from OCO-2 inversions are in better agreement with bottom up estimates (e.g. Table 1 in Williams et al. (2007)) than the prior and *in situ* experiment flux estimates. Of further note is the similarity of flux seasonality in these regions derived from OCO-2 retrievals to land surface models employing prognostic phenology (i.e. ORCHIDEE and SiB4, which are used as prior fluxes by the CAMS and CSU models as described in Appendix A). These two factors indicate that the OCO-2 inferred fluxes may not be driven by retrieval biases.

### 4.5 Evaluation Against Independent Data

The fluxes discussed in the previous sections indicate different signals present in the OCO-2 land retrievals than from the global network of *in situ* measurements, particularly in the tropics. Given the scarcity of *in situ* measurements in these regions, particularly when compared to the number of OCO-2 soundings, this is not surprising. However, perennial cloudiness in the Tropics, as well as aerosols arising from biomass burning and dust, both reduce the number of OCO-2 soundings and potentially induce biases in the remaining data. These facts leave the question of accuracy in the posterior fluxes unanswered. In order to explore this question, we evaluate the posterior fluxes by sampling the resultant concentrations for comparison with TCCON and aircraft measurements.

#### 4.5.1 TCCON

All modelers sampled their posterior concentration fields at TCCON retrieval locations and times to compare directly to the TCCON dataset as available during the full period starting January 1, 2015 and ending April 1, 2017. Not all sites have the same length of record due to latency in the release of quality controlled data. Time series of simulated and retrieved $X_{CO_2}$ at TCCON sites are shown in Figures C2-C4, from which the length of the available records for each site can be seen.

Figure 8 depicts the overall error statistics for each model by site and data constraint. The model concentrations are sampled for each 30 minute average TCCON retrieval, as described in the experimental design, and then subtracted from the TCCON values to calculate statistics. For comparison to OCO-2 retrievals, available 10s retrievals from OCO-2, using a 5 degree latitude and longitude geometric coincidence criteria, were averaged and compared to TCCON observations occurring within one hour of the overpass time, in much the same way that a coarse global transport model would be sampled for this purpose. For the LN and LG experiments in the middle and bottom rows of Figure 8, error statistics for co-located OCO-2 observations are also displayed in the first column of the panel to give a sense of the correlation between the OCO-2 retrievals and the resulting modeled concentrations at each TCCON site. Of note is the strong correlation between OCO-2 mismatches with TCCON and the posterior simulated concentration mismatches with TCON. For example, the OCO-

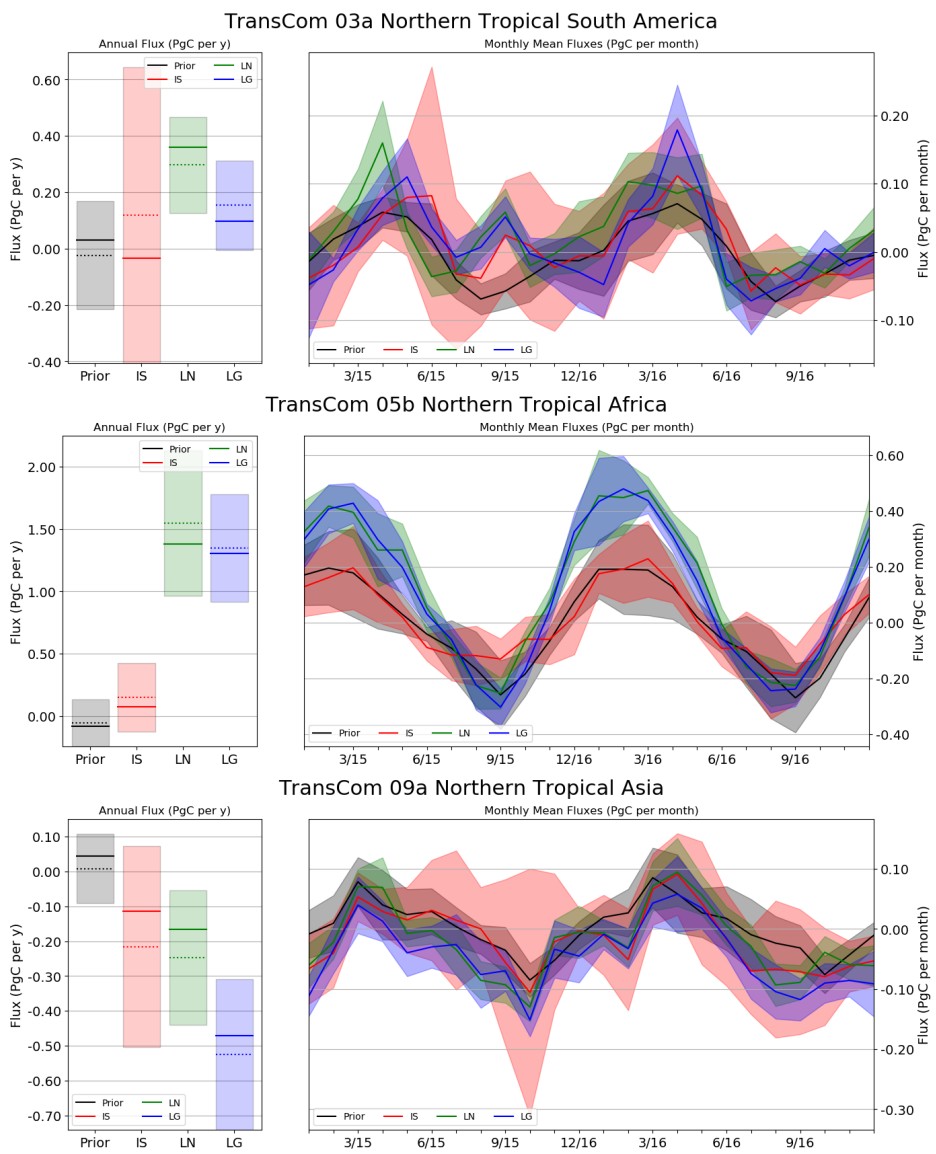

**Figure 6.** As in the right column of Figure 3, but for selected terrestrial regions in the northern tropics on different continents. Among the three continents, satellite-derived posterior fluxes differ substantially from the prior and *in situ* constrained fluxes only in northern tropical Africa, where the outgassing in the dry season is about double in magnitude. The phasing in the posterior fluxes is also different in South Tropical Africa, where the peak outgassing is shifted later in the year by a few months.

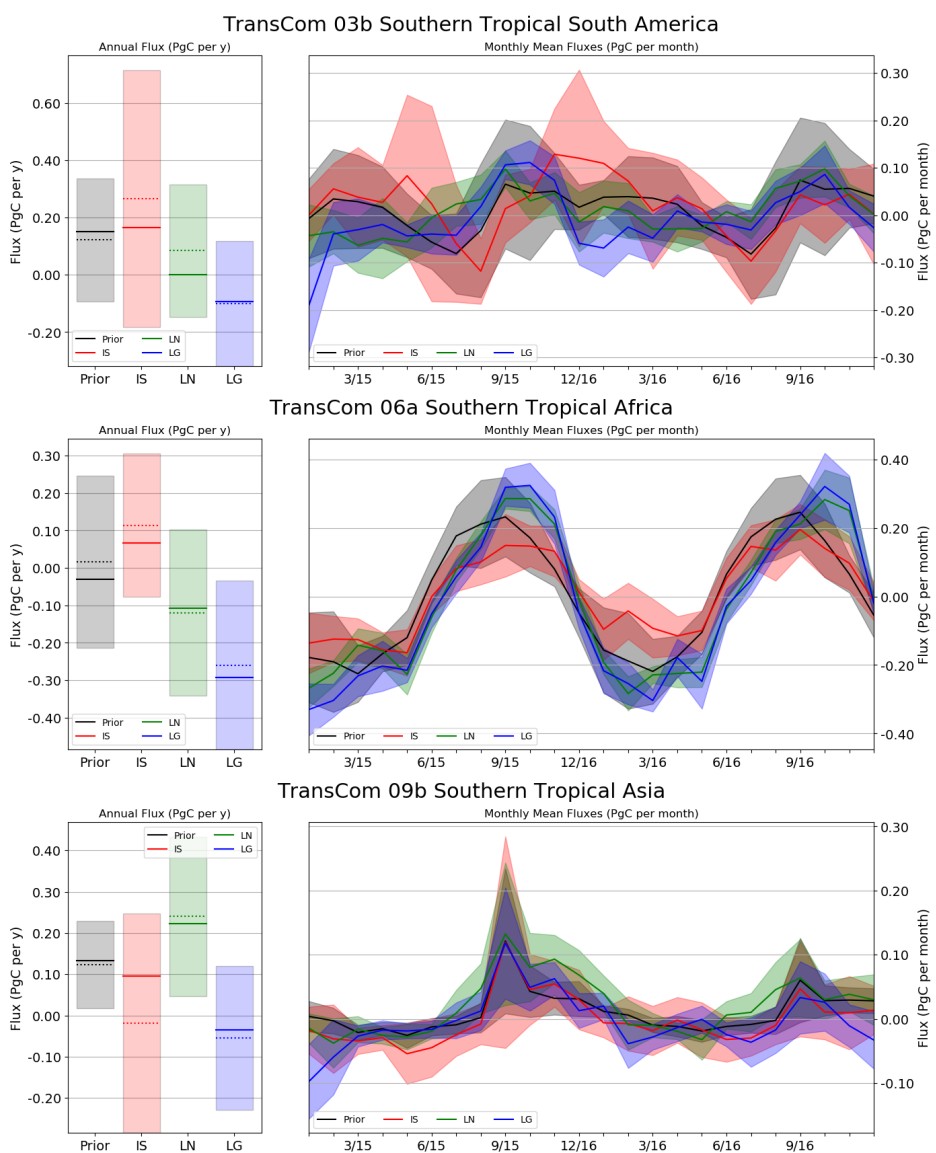

**Figure 7.** As in the right column of Figure 3, but for selected terrestrial regions in the southern tropics on different continents. Among the three continents, satellite-derived posterior fluxes differ substantially from the prior and *in situ* constrained fluxes only in tropical Africa, where the peak outgassing is shifted later in the year by a few months.

2 land nadir retrievals are biased high relative to most TCCON sites, in line with estimates from
Chatterjee et al. (in preparation), and the LN inversion simulated concentrations show a similar high
bias across models. The European TCCON sites show a consistent pattern, in which all model concentrations are biased high. This indicates an issue with representativeness of coarse global transport
models at these sites or with the accuracy of the TCCON retrievals, though no evidence for the latter
has been presented in the literature. Another similarity across the results is the strong difference between residuals for the Dryden and Caltech sites, which are located very close to one another. This is
due to the highly local nature of these observations and the relatively broad coincidence criteria used
in the comparison. Coarse models are unable to simulate all of the variability at these sites. Caltech
in particular is highly influenced by the Los Angeles basin, while Dryden, though geographically
close to Caltech, is separated from the basin by mountains and thus samples the relatively clean
environment outside the basin (Kort et al., 2012; Schwandner et al., 2017). The high bias at Dryden
is likely due in part to models simulating conditions from inside the Los Angeles basin, and the low
bias at Caltech due to models simulating some of the cleaner air north of the basin. The challenges of
comparing point data to model grid cell concentrations highlights that representativeness and model
resolution are key issues for using TCCON and other data sets to evaluate model results.

There are four TCCON sites in the Tropics: Manaus, Ascension Island, Reunion Island, and Darwin. These sites all have different seasonal flow patterns that result in varying upwind source regions
that may make it difficult to use TCCON column data to validate inverse model fluxes. The time series of residuals are shown in Figure C4. LN posterior concentrations have a similar high bias for
all four sites. LG posterior samples are biased high at Ascension, low at Reunion, with a seasonally
varying bias sign at Darwin. The biases in the IS posterior concentrations are scattered around zero
at Darwin and Reunion, though they are uniformly high at Ascension. Correlating these residuals
to flux patterns is difficult for the reasons listed above. For example, the LG and LN posterior ensembles have similar ensemble mean monthly fluxes in the North and South Tropics as zonal bands
as well as the land regions that make up these zonal bands, but time series comparisons of each to
TCCON do not demonstrate this.

### 4.5.2 Surface and Aircraft *In Situ* Observations

The posterior concentrations were sampled at the locations and times of the surface sites shown in
Figure 1 as well as the CONTRAIL flights for 2015 and the available ATom and ORCAS flight
campaigns in the time period of the experiments, i.e. 2016-2017. The results of the comparisons are
shown in Figure 9, including both bias and error standard deviation for different latitudes (along the
horizontal axis) and altitudes by row.

As depicted in the upper left panel of Figure 9, the IS posterior concentrations compare well with
the PBL measurements; this is expected as they assimilate these data to optimize the surface fluxes.
However, LN, LG and the prior all have a positive bias in the in the northern extratropics, indicating

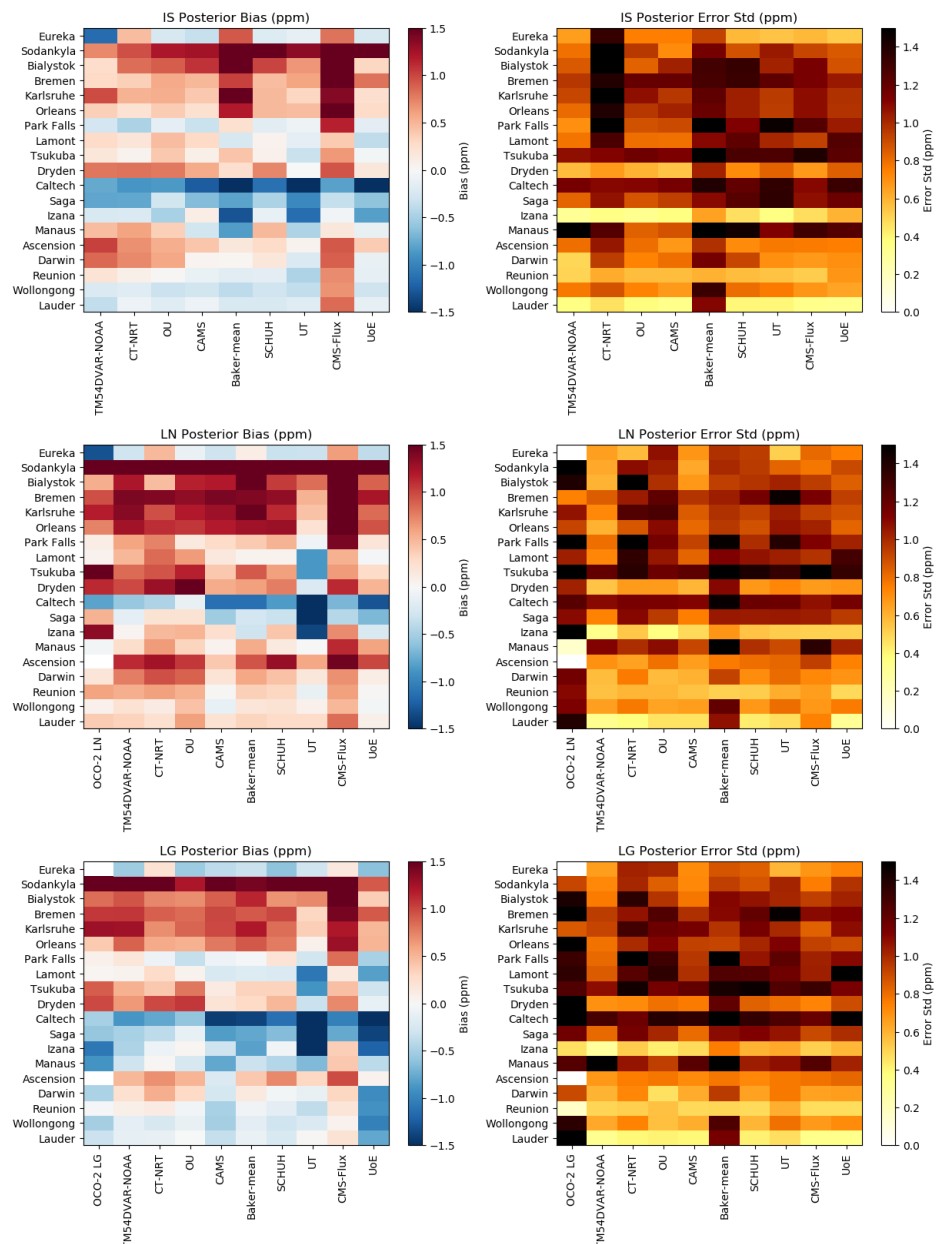

**Figure 8.** Bias (left) and standard deviation (right) for all TCCON sites by model (ordered by latitude). Statistics are computed from all residuals: simulated - retrieved $X_{CO_2}$. For the OCO-2 LN (middle) and OCO-2 LG (bottom) statistics, the first column depicts the statistics for the residuals between collocated OCO-2 10s values and TCCON retrievals. Of note is the correlation between the bias in the OCO-2 retrievals and the resultant bias in the posterior concentrations. In general, LN experiment posterior samples show a high bias relative to TCCON at all except a few sites. IS and LG show similar biases over most of the sites. According to O'Dell et al. (2018b), land nadir observations are biased high by about 0.5ppm relative to land glint observations. The LN experiment also has a larger standard deviation at most TCCON sites than the IS and LG experiments. Of particular interest are the various European sites, for which all models and data constraints show a high bias relative to TCCON.

too much overall $CO_2$ in that region at all three atmospheric layers. Interestingly, above the PBL
      in the tropics, LN has the lowest bias of the three experiments, though with the important caveat
      that this comparison is driven totally by two seasons (boreal winter and spring) of ATom aircraft
      measurements with flights in the Atlantic and the Pacific. Thus, we cannot draw the conclusion that
      the enhanced tropical outgassing in the northern tropics in the OCO-2 constrained fluxes is correct,
particularly since LG posterior samples resemble the IS posterior samples more so than LN in the
      tropics, while the LG fluxes are more in line with LN. Lastly, none of the observational constraints
      improves the overall simulated variability in atmospheric concentrations relative to the observations
      in any of the three atmospheric layers presented, at all latitudes, as shown in the right column of
      Figure 9. This is likely due to the coarse spatial resolution of the models included in this study.

It is tempting to draw conclusions about surface fluxes from these comparisons with independent
      data. However, the general sparseness of these samples in space and/or time as well as the seeming
      lack of correspondence between the posterior flux differences across experiments and their posterior
      concentrations across experiments makes this difficult to do. For example, as mentioned above, LN
      and LG posterior monthly fluxes are similar in the tropics, but the posterior concentrations of LG
compare better with IS than LN in the tropics in the mean. A detailed examination of the goodness of
      fit of the experimental posterior concentrations with each observational data set is beyond the scope
      of this work.

## 5   Discussion

      We have used a suite of atmospheric inverse models to analyze the OCO-2 $X_{CO_2}$ retrieval data to
identify $CO_2$ flux signals that stand out above the noise of transport model error and inversion as-
      sumption differences. The OCO-2 retrievals for different viewing modes (LN,LG) were assimilated
      in separate experiments given the differences between the signals present in each, as detailed in
      Chatterjee et al. (in preparation). We have presented these flux results starting at the global scale,
      then moving to broad zonal results, and focusing finally on results at the continental scale; at this
finest scale, we present results for the land regions only, since we do not expect the satellite data
      taken over land to provide a strong constraint on the ocean fluxes. The inversions point to several
      areas where the OCO-2 data drive robust differences from our prior flux estimates, in some cases
      differing from the results given by the *in situ* data and in other cases showing agreement.

          First, we note that even with the global coverage provided by OCO-2, we do not see a reduction in
ensemble spread, the possibility of which is alluded to in the introduction. Given the work shown in
      Schuh et al. (2019) and Basu et al. (2018), we suspect that this is at least partially driven by transport
      differences. There are likely residual regional biases in the OCO-2 data themselves also, and the
      way they manifest in the fluxes in going to be highly dependent on the transport model and inversion
      framework.

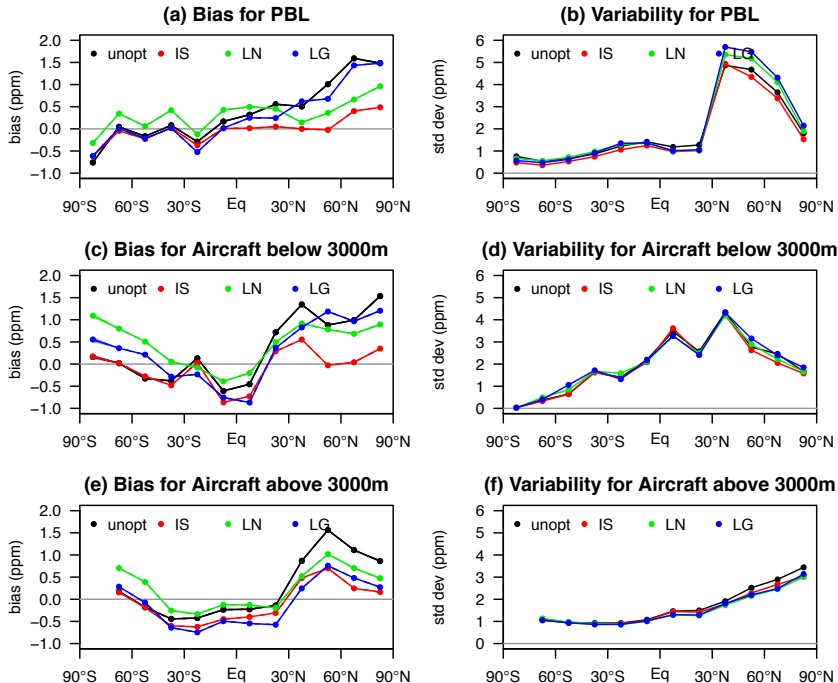

**Figure 9.** Comparisons between *in situ* measurements and unoptimized and optimized posterior concentrations using different observational constraints. The left column depicts the overall bias and the right column the standard deviation of the errors, each plotted against latitude. The rows distinguish between measurements in the PBL (top row), aircraft below 3000m ASL (middle row), and aircraft above 3000m (bottom row). Aircraft measurements include the NOAA light aircraft profiles (Sweeney et al., 2015), CONTRAIL flask and analyzer data as well as observations from the ORCAS and ATom campaigns. The PBL measurements were assimilated in the IS experiments, which is apparent given the very low bias in the top row for the red curve. The IS experiments exhibit the smallest bias throughout the atmosphere in the northern extratropics and above the PBL in the southern extratropics (largely driven by ORCAS data), while the LN posterior concentrations have the lowest bias above the PBL in the tropics. LG posterior concentrations in general follow the unoptimized concentrations, with a slight negative or positive shift that depends on latitude and altitude. None of the observational constraints improves the match to the variability in the observations much over the prior mean.

In the northern extra-tropics, the most robust signal in the inversion results is the phase adjustment of the seasonal cycle of net ecosystem exchange on land, as well as a deeper maximum summertime drawdown relative to the prior mean fluxes. Peak carbon draw down appeared approximately a month early than expected, as did the onset of net positive fluxes in the early fall. In future work, it would be useful to see how these shifts in NEE agree with the solar induced fluorescence products that are now being produced by OCO-2 (Frankenberg et al., 2014; Sun et al., 2017) and the TROPOspheric Monitoring Instrument (TROPOMI). In the southern extra-tropics, we did not find significant differences from the *a priori* fluxes, probably because the limited amount of land data available that far south precluded inference about the fluxes there. The OCO-2 data hint at a somewhat higher-amplitude seasonal cycle in the global ocean fluxes than we had in our priors, but the experiments of Basu et al. (2018) caution us that ocean fluxes inferred from land data only may be particularly susceptible to sampling bias, transport errors, over-reliance on prior fluxes, and the inability of coarse models to constrain land and ocean fluxes separately.

As mentioned previously, a key promise of satellite data is to provide new information relative to the global *in situ* network in the tropics, where the *in situ* data provide a minimal constraint, and that is in fact the case: the OCO-2 data imply a significantly larger seasonal cycle in the tropics than given in our prior or given by the *in situ* data, in terms of the land+ocean flux total. This greater seasonality is driven by the land fluxes, and most of it occurs in Africa, both north and south of the equator. The strongest of these deviations is evident in northern Africa, where annual net fluxes of carbon were $1.5 \pm 0.6$ PgC per year for LN and $0.8 \pm 0.6$ PgC per year for LG (carbon efflux to atmosphere). The seasonality of fluxes in this area was also much stronger than in many of the prior land fluxes, which in our experiments arise from terrestrial ecosystem models. For example, in Tropical North Africa, the LN and LG mean seasonal amplitude (i.e. max minus min flux) was about 1 PgC per month, while in the prior fluxes, the amplitude was about 0.4 PgC per month In particular, the positive adjustment in carbon fluxes from November to June time frame were the driving force behind posterior adjustments to both annual fluxes and seasonal amplitude. While this topic is beyond the scope and focus of this paper, we feel obliged to discuss possible candidate processes that might contribute to what we see in North Africa. The positive flux adjustments we obtain there fall squarely within the strong local dry season, raising stronger carbon inputs from fire as an obvious possibility. However, fires are imposed within most of the modeling systems and the likelihood of fire emissions being wrong by 1 PgC or more seems slim, which implies that fires alone cannot explain the results. Liu et al. (2017) found that respiration was an important part of the anomalous efflux (relative to a La Niña period) from this region during the time period of interest, which offers a potential explanation. Northern Africa is an area with large expanses of high surface albedo and aerosols due to wind and dust sources. Reasonable effort has been made to evaluate the potential biases in the area by running atmospheric inversions with simulated biases in areas of concern (not shown) as well as analysis of downwind TCCON sites such as Ascension Island. With

no clear indicators of bias and given the sparseness and representativeness of the available evaluation data, we cannot falsify either the IS-constrained tropical fluxes or the satellite-constrained fluxes, despite the large difference between them. Therefore, we must move forward with the hypothesis that this signal may be valid and is tied to variations in either respiration, photosynthesis, or both.

Next, we point to the observation made in Section 4.4 where the suite of inversion results for Northern Tropical Africa tend to move toward the fluxes from the SiB4 and ORCHIDEE prognostic biosphere models. An analysis of the SiB4 prior fluxes indicate very strong seasonal flux signals from C4 grasslands in the region. Grasslands have large quick-turnover carbon pools and thus it is not surprising that respiration and photosynthesis are strongly correlated seasonally. There are also strong respiration and photosynthesis fluxes in deciduous and evergreen broad-leaf plant types in this area although the longer turnover wood pools imply that the seasonality in the NEE for this vegetation is likely driven more strongly by photosynthesis. Grasslands have historically been very difficult to model with NDVI/EVI driven diagnostic biosphere models such as CASA and thus seem a natural candidate to explain higher posterior NEE amplitude. The larger amplification in the dry season could also point to more subtle reductions in photosynthesis across forested regions not being captured by the diagnostic models, where there is often difficulty due to the saturation of vegetation indexes such as NDVI. The posterior adjustments from the models seem to imply a stronger annual sources and a stronger seasonal cycle, likely implying some combination of effects from both forests and grasslands.

We also not the difficulty in constraining ocean fluxes with only LN data and in partitioning land and ocean fluxes due to inconsistencies between land nadir and ocean glint modes (Basu et al., 2018). Ocean glint retrievals in v7 of the data were unusable due to systematic biases discovered during this exercise. In light of this, several improvements were made in Version 8 (O'Dell et al., 2018a), and retained in Version 9 (Kiel et al., 2018) of the OCO-2 retrievals and we hope will make the ocean glint data more informative in the next round of experiments. The continued difficulty of using data with biases between different modes (e.g. ocean glint vs land nadir) emphasizes the potential value of "online" bias correction methods which allow for the post-hoc OCO-2 bias correction to be performed in a consistent fashion within the atmospheric inversion framework, as well as alternate methods of using information on the $CO_2$ vertical information present in the retrievals.

## 6   Conclusions

Satellite retrievals have tremendous potential for constraining surface fluxes of $CO_2$ (Rayner and O'Brien, 2001). In this study, we employ an ensemble of inversion models with different assumptions to estimate surface $CO_2$ fluxes in 2015 and 2016, and their uncertainties. We find that OCO-2 retrievals inform fluxes that agree at global scales with those of *in situ* data. Furthermore, agreement is found where both satellite and *in situ* data are dense enough to provide sufficient constraint.

The inferred fluxes differ significantly in the tropics, where the satellite retrievals suggest a much stronger seasonal cycle than the *in situ* measurements over most of the zone, and in particular a much stronger outgassing from the Northern Tropics, with the main differences occurring in Africa. Ocean fluxes generally remain close to the prior in all experiments.

Evaluating this new flux information is a difficult task. The TCCON retrievals suggest that the tropical outgassing in the LN experiments is too large, but this is weakened by the site dependence of the errors in these TCCON comparisons. PBL and aircraft observations lead to different conclusions, but again these are sparse and potentially do not capture the influence of fluxes from the regions in question.

Despite the difficulties in evaluating the OCO-2 derived flux estimates obtained here, the comparison to more traditional *in situ*-based estimates has been illuminating. The satellite results have exposed the sensitivity of the *in situ* results to the transport used, especially the vertical transport: spread in the *in situ* results is largest over tropical land regions, and the satellite results provide their most robust new insight into the global carbon cycle, especially in terms of the magnitude and timing of the seasonal cycle of flux. This process of questioning old results and testing the new results will continue as the satellite data are used in new ways. The impact of using vertical information from the satellite retrievals (instead of just the straight vertical mean given by $X_{CO_2}$) is a notable area of on-going research: the bias correction of the OCO-2 retrievals with respect to TCCON $X_{CO_2}$ should be expected to change considerably as the information from the satellites closer to the surface is emphasized more.

In the future, the analysis shown here will be repeated with updated OCO-2 retrievals, and new analyses performed for a longer period that includes 2017-on. The new Version 9 OCO-2 retrievals should have lower overall biases compared to Version 7 used for these experiments. In particular, the ocean glint retrievals should be significantly improved, due to the inclusion of aerosol dynamics that are expected to eliminate the bias in the high southern latitudes (O'Dell et al., 2018b). This will provide an exciting opportunity for constraining ocean fluxes. Additionally, an updated ACOS GOSAT product for the entire data record is due to be released in 2019, and the comparison of OCO-2 constrained fluxes with the much longer GOSAT record is critical for understanding the long term behavior of the tropical carbon cycle.

*Acknowledgements.* SC is funded by NASA grant number NNX15AJ37G. The work of FC has been funded by the Copernicus Atmosphere Monitoring Service, implemented by the European Centre for Medium-Range Weather Forecasts (ECMWF) on behalf of the European Commission. LF and PIP acknowledge support from the UK National Centre for Earth Observation (NCEO). NERC provides national capability funding to NCEO (grant #PR140015). AS is funded by NASA grant number NNX15AG93G. Part of the research described in this paper was carried out at the Jet Propulsion Laboratory, California Institute of Technology, under a contract with the National Aeronautics and Space Administration. The National Center for Atmospheric Re-

| Group | Transport Model | Spatial Resolution | DA Method | Prior: Land Bio | Prior: Ocean | Prior: Fire | Reference |
|---|---|---|---|---|---|---|---|
| OU | TM5 | $4° \times 6°$ | 4DVAR | CT-NRT Unopt | CT-NRT Unopt | CT-NRT Unopt | Crowell et al. (2018) |
| CT-NRT | TM5 | $2° \times 3°/1° \times 1°$ | EnKF | CT2016 Opt Clim | CT2016 Opt Clim | CT2016 Clim | Peters et al. (2007)[a] |
| CAMS | LMDZ | $1.875° \times 3.75°$ | Variational | ORCHIDEE Clim | Landschuetzer et al. | GFAS | Chevallier et al. (2005)[b] |
| Schuh | GEOS-Chem | $1° \times 1°$ | Bayesian Synthesis | SiB4/MERRA | CT2015 Opt Clim | None | |
| TM54DVAR-NOAA | TM5 | $2° \times 3°$ | 4DVAR | SiB-CASA | CT2015 Opt Clim | GFEDv4 | Basu et al. (2013) |
| UoE | GEOS-Chem | $4° \times 5°$ | EnKF | CASA | Takahashi et al (2009) | GFEDv4 | Feng et al. (2009),Feng et al. (2016) |
| UT | GEOS-Chem | $4° \times 5°$ | 4DVAR | BEPS | Takahashi et al. (2009) | GFEDv4 | Deng and Chen (2011a) |
| CMS-Flux | GEOS-Chem | $4° \times 5°$ | 4DVAR | CASA-GFEDv3 | ECCO2-Darwin | GFEDv3 | Liu et al. (2014a) |
| Baker | PCTM | $6.7° \times 6.7°$ | 4DVAR | CASA-GFEDv3 | Takahashi/... | GFEDv3 | Baker et al. (2010) |

**Table 2.** Key model parameters for each of the members of the OCO-2 MIP. More information and references are contained in Section A.

[a] with updates documented at http://carbontracker.noaa.gov

[b] with updates documented at https://atmosphere.copernicus.eu/sites/default/files/2018-10/CAMS73_2015SC3_D73.1.4.2-1979-2017-v1_201807_v1-1.pdf

search is sponsored by the National Science Foundation. The work of KM was funded by NASA grant number NNX16AAL92A.

The authors would like to thank the institutions that provide data from the TCCON network as well as the providers of the *in situ* observations.

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

## Appendix A: Model Information

This section contains the description of each modeling framework, including key references that describe the methodology.

### A1   Baker

This set of results uses the variational carbon data assimilation system of Baker et al. (2006a), which solves for weekly corrections to a set of net surface $CO_2$ fluxes on the lat/lon grid of an underlying transport model. This transport model is the parameterized chemical transport model (PCTM) of Kawa et al. (2004), driven by meterological and mixing parameters from the MERRA-2 reanalysis (Bosilovich et al., 2017). The MERRA-2 fields are coarsened from their original 0.5°x0.625° (lat/lon) resolution on 72 vertical levels to 40 vertical levels at 2.0°x2.5° resolution for forward runs of the prior fluxes and 6.67°x6.67° resolution for the assimilation of the measurements. Prior fluxes included gross primary productivity (GPP), autotrophic and heterotrophic respiration, wildfire, and

biofuel burning fluxes from the CASA land biosphere model (van der Werf et al., 2004), as well as air-sea $CO_2$ fluxes from a suite of four ocean models: NOBM, Takahashi et al. (2009), Landschützer et al. (2015), and the same Landschuetzer fluxes with a southern ocean sink of 0.95 PgC/yr added on, with a separate set of inversions performed for each of the four ocean priors. For each of the four sets of priors, a multiple of the CASA global respiration fluxes plus a global offset are solved for to force the prior to match the 2008-2015 trend at NOAA's Mauna Loa flask site. The net flux for these four sets of priors are run forward through PCTM at 2.0°x2.5° (lat/lon) resolution for 2008-2018, starting from a realistic initial 3-D $CO_2$ field; the resulting $CO_2$ fields are sampled at the times and places of the *in situ*, TCCON, and OCO-2 measurement locations used here with a suitable vertical weighting; and the mismatches to the actual measurements used to estimate corrections to the prior fluxes using the variational method running PCTM at 6.67°x6.67° resolution. Separate assimilations are done starting from each of the four sets of priors, and the average fluxes from these four cases are used here. The prior flux uncertainties used are those from (Baker et al., 2006b).

## A2 CAMS

CAMS uses the $CO_2$ inversion system of the Copernicus Atmosphere Monitoring Service (http://atmosphere.copernicus.eu/), called PyVAR-CO2 (Chevallier et al., 2005, 2010a, 2017) directly adapted to the OCO-2 MIP protocol. It solves the Bayesian inference problem by the minimization of a cost function using the Lanczos version of the conjugate gradient algorithm (Fisher 1998, Desroziers and Berre 2012).

The transport model in the configuration of PyVAR-CO2 for this study is the global general circulation model LMDZ in its version LMDZ3 (Locatelli and et al, 2015), that uses the deep convection model of Tiedtke (1989). This version has a regular horizontal resolution of 3.75° in longitude and 1.875° in latitude, with 39 hybrid layers in the vertical. It is nudged towards the ERA-Interim re-analysis (Dee, 2011). Note that the official CAMS releases use a different, more computationally expensive, convection model (Emanuel, 1991). For the computational efficiency of the variational approach, PyVAR-CO2 uses the tangent-linear and adjoint codes of LMDZ.

The inferred fluxes are estimated in each horizontal grid point of the transport model with a temporal resolution of 8 days, separately for day-time and night-time. The state vector of the inversion system is therefore made of a succession of global maps with 9,200 grid points. Per month it gathers 73,700 variables (four day-time maps and four night-time maps). It also includes a map of the total $CO_2$ columns at the initial time step of the inversion window in order to account for the uncertainty in the initial state of $CO_2$.

The prior values of the fluxes combine estimates of monthly ocean fluxes (Landschützer et al., 2015), 3-hourly (when available) or monthly biomass burning emissions (GFAS, http://atmosphere.copernicus.eu/) and climatological 3-hourly biosphere-atmosphere fluxes taken as the 1989-2010 mean of a simulation of the ORganizing Carbon and Hydrology In Dynamic EcosystEms model

(ORCHIDEE, (Krinner et al., 2005)), version 1.9.5.2. The mass of carbon emitted annually during specific fire events is compensated here by the same annual flux of opposite sign representing the re-growth of burnt vegetation, which is distributed regularly throughout the year. The gridded prior fluxes exhibit 3-hourly variations, but their inter-annual variations over land are only caused by anthropogenic emissions.

Over land, the errors of the prior biosphere-atmosphere fluxes are assumed to dominate the error budget and the covariances are constrained by an analysis of mismatches with *in situ* flux measurements (Chevallier et al., 2006, 2012) : temporal correlations on daily mean Net Carbon Exchange (NEE) errors decay exponentially with a length of one month, but night-time errors are assumed to be uncorrelated with daytime errors; spatial correlations decay exponentially with a length of 500 km; standard deviations are set to 0.8 times the climatological daily-varying heterotrophic respiration flux simulated by ORCHIDEE with a ceiling of 4 gC per m$^2$ per day. Over a full year, the total 1-sigma uncertainty for the prior land fluxes amounts to about 3.0 GtC per year. The error statistics for the open ocean correspond to a global air-sea flux uncertainty about 0.5 GtC per year and are defined as follows: temporal correlations decay exponentially with a length of one month; unlike land, daytime and night-time flux errors are fully correlated; spatial correlations follow an e-folding length of 1000 km; standard deviations are set to 0.1 gC per m2 per day. Land and ocean flux errors are not correlated.

## A3    CMS-Flux

CMS-Flux, where CMS stands for Carbon Monitoring System, optimizes monthly terrestrial biosphere and ocean carbon fluxes using 4D-Var inversion approach with GEOS-Chem adjoint model (Liu et al., 2014b). The model is run at 4° (lat) x 5° (lon) spatial resolution driven by GEOS-FP meteorology. The prior biosphere fluxes are based on CASA-GFED3 (van der Werf et al., 2004), while ocean carbon fluxes are from ECCO2-Darwin (Dutkiewicz et al., 2009; Follows et al., 2007; Follows and Dutkiewicz, 2011). Both ocean and biosphere fluxes are 3 hourly. We assumed no correlation in prior flux uncertainties in both space and time.

## A4    CSU-Schuh

We use a Bayesian technique with SiB4 as the carbon flux prior model for respiration and gross primary production (GPP). SiB4 is an integration of heterogeneous land-atmosphere fluxes, environmentally responsive prognostic phenology, dynamic carbon allocation, and cascading carbon pools from live biomass to surface litter to soil organic matter. Rather than relying on satellite data for the vegetation state, SiB4 brings together biological phenology, plant physiology, and ecosystem biogeochemistry to fully simulate the terrestrial carbon cycle, predicting consistent energy exchanges, carbon fluxes and carbon pools. To capture vegetation-specific phenology and biological processes, SiB4 uses twenty-four plant functional types (PFTs), including three specific crops (maize, soybean

and winter wheat). For this work, SiB4 fluxes were provided at 1° x 1° degree resolution. Each 1° x 1° box could consist of up to 24 PFTs, responding in a joint way to the atmosphere. Thus there is no effective "round off" error from using a single dominant PFT or biome on a coarse land surface grid.

We use a conceptually simple inversion framework with the goal of providing optimized $CO_2$ fluxes for plant functional types (PFTs) on continental scales. In particular, for each of 25 possible PFTs, and each of 11 Transcom land regions, we solve for $\beta$, the amplitudes of the Fourier harmonics, in the following equations:

$$Opt_{GPP}(DOY) = Prior_{GPP}(DOY) * \left( \beta_0^c + \sum_{k=1}^{3} \beta_k^s \sin DOY/365 * 2\pi k + \sum_{k=1}^{3} \beta_k^c \cos DOY/365 * 2\pi k \right)$$

$$Opt_{RESP}(DOY) = Prior_{RESP}(DOY) * \left( \beta_0^c + \sum_{k=1}^{3} \beta_k^s \sin DOY/365 * 2\pi k + \sum_{k=1}^{3} \beta_k^c \cos DOY/365 * 2\pi k \right)$$

This framework optimizes the seven $\beta$ coefficients shown above for each of up to 25 PFTs for each of 11 Transcom Regions for GPP and respiration (separately) for a total of up to 7\*25\*11\*2 = 3696 parameters. To illustrate this, two trivial univariate examples are presented for GPP in the Missouri Ozarks Ameriflux site and total respiration in the Howland Forest Ameriflux site in Maine. Ocean regions are divided into 30 regions according to Jacobson et al. (2007) and solved for in a similar fashion to land but with only 2 harmonics.

In practice, each of the stochastically fixed coefficients to the betas are run through GeosCHEM v11 as individual pulses. We only need to run each of these pulses once and it is not necessary to split up the pulse in time (e.g. months) because this is what one gets from the posterior reconstruction of the flux signals. The number of harmonics determine the highest frequency flux signals to be expected. With three harmonics, we expect to be able to recover seasonal corrections on time scales down to about 2 months. Each pulse provides a vector of sensitivities of the observations to that particular pulse. We then concatenate these vectors into a large Jacobian (sensitivity matrix) and solve for the regression coefficients

## A5  CT-NRT

CarbonTracker Near-Real Time (CT-NRT) is an extension of the formal CarbonTracker $CO_2$ analysis system, designed to bridge the gap between annual updates of NOAA's formal CarbonTracker product. It extends model results beyond the most recent CarbonTracker release until the end of available ERA-interim meteorology needed to drive its transport model, TM5. The release of CT-NRT used in this study, CT-NRT.v2017, was initialized in September 2014 from the CT2016 release of CarbonTracker (Peters et al., 2007, with updates documented at http://carbontracker.noaa.gov). CT-NRT uses a unique set of flux priors, derived from the optimized fluxes of CT2016. The 2001-2015 climatology of these optimized terrestrial fluxes is augmented with a statistical model of flux anomalies,

also derived from CT2015 results. Ocean and wildfire prior fluxes are set to the seasonally-varying climatology of optimized CT2016 fluxes without interannual variability. This prior not only has a long-term mean terrestrial sink, but also attempts to represent interannual variability in land $CO_2$ flux due to anomalies of temperature, precipitation, and solar insolation. This prior was developed to mitigate the smaller number of *in situ* $CO_2$ measurements available for assimilation in near-real time, as it is presumably less biased than the standard CarbonTracker prior with its small land sink.

### A6  TM54DVAR-NOAA

The TM5 4DVAR system is a Bayesian inverse modeling framework that infers surface fluxes of a tracer given measured tracer mole fractions in the atmosphere (Meirink et al., 2008). It uses the TM5 atmospheric chemistry transport model to connect atmospheric measurements with surface fluxes (Krol et al., 2005) . TM5 and its adjoint are used for a variational estimate of surface fluxes. For this work, we ran TM5 globally at 3 ° lon x 2 ° lat with 25 vertical layers. We used TM5 4DVAR to solve for terrestrial and oceanic $CO_2$ fluxes, with fixed fossil fuel fluxes described elsewhere in this manuscript. Prior oceanic fluxes were constructed from a climatological average of CT2015 oceanic flux estimates. Terrestrial $CO_2$ fluxes – the sum of net ecosystem exchange and fire fluxes – were taken from SiB CASA GFED 4 (van der Velde et al., 2013). The uncertainty on the terrestrial fluxes were fixed to be 0.5 x heterotrophic respiration from SiB CASA, while the uncertainty on oceanic fluxes was fixed at 1.57 times the absolute flux at each grid cell and time step. The uncertainty of the prior flux is assumed to have exponential spatio-temporal correlation, with length and time scales of 1000 km and 3 weeks for the oceanic component and 250 km and 1 week for the terrestrial component. OCO-2 retrievals assimilated are described elsewhere in this document, while the *in situ* $CO_2$ measurements assimilated were identical to the set used by CT-NRT.

### A7  University of Oklahoma (OU)

The OU results utilize the same model and data assimilation framework as the TM54DVAR-NOAA group, but with different inputs. The OU experiments utilize the CT-NRT unoptimized prior emissions, and uncertainties derived from different climatological fluxes. The initial conditions are provided by CarbonTracker, and the model constrains monthly 6° by 4° emissions from March 1, 2014 though April 1, 2017. The OU system uses the same prior fluxes as CT-NRT, and so provides a measure of the contribution of the data assimilation framework, prior uncertainties, and spatial resolution to posterior emissions. Conversely, the OU experiment provides the impact of prior emissions and uncertainties and spatial resolution relative to the TM54DVAR-NOAA results.

### A8  University of Edinburgh (UoE)

The UoE inversions are based on an Ensemble Kalman Filter framework (Feng et al., 2009, 2016) for inferring surface $CO_2$ fluxes by optimally fitting model simulation with the in-situ or space-based

measurements of atmospheric $CO_2$ concentrations. We use the global 3D chemistry transport model
(CTM) GEOS-Chem of version 9.02 to simulate model $CO_2$ concentrations at a horizontal resolution
of 4° (latitude) by 5° (longitude), driven by the GEOS-FP meteorological analyses from the Global
Modeling and Assimilation Office Global Circulation Model based at NASA Goddard Space Flight
Center.

The prior surface fluxes are taken from existing emission inventories, including: 1) monthly
biomass burning emission (GFEDv4.0, Van der Werf et al. (2010));and 2), monthly climatologi-
cal ocean fluxes (Takahashi et al., 2009); and 4) three-hourly terrestrial biosphere fluxes (CASA,
(Olsen and Randerson, 2004)). We assume a 60% uncertainty for land monthly fluxes, and 40% for
oceanic fluxes. Errors for land (ocean) prior fluxes are also assumed to be correlated with each other
with a correlation length of 500 (800) km. By optimally fitting model simulation with observations,
we infer monthly $CO_2$ fluxes over 792 geographic regions (475 land regions and 317 ocean regions),
compared to the 199 global regions used in our previous experiments (Feng et al., 2009).

**A9   University of Toronto (UT)**

UT results employ the GEOS-Chem (http://geos-chem.org) global three-dimensional chemical trans-
port model, driven by assimilated meteorological observations from the Goddard Earth Observing
System version 5 of the NASA Global Modeling Assimilation Office. The model configuration is the
same as that used in Deng et al. (2016). The resolution of the model is 4° x 5°, with 47 vertical lev-
els extending from the surface to 0.01 hPa. The assimilation is carried out using a four-dimensional
variational (4D-Var) approach (Henze et al., 2007).

The *a priori* $CO_2$ flux inventories are the following. For biomass burning, we used monthly emis-
sions from the Global Fire Emissions Database version 4 (urlhttp://www.globalfiredata.org/). The
atmosphere-ocean flux of $CO_2$ is based on the monthly climatology of Takahashi et al. (2009). For
the biospheric flux of $CO_2$, we use 3-hourly fluxes from the Boreal Ecosystem Productivity Simula-
tor (Chen et al., 2012) . As in Deng et al. (2014), it is assumed that the annual terrestrial ecosystem
exchange is neutral in each grid box Deng and Chen (2011b). Although the temporal resolution for
the terrestrial ecosystem exchange is 3 h, the optimized scaling factors are estimated with a monthly
temporal resolution.

Diagonal priori error covariance matrix was used and it is assumed (Deng et al., 2016) that the
1-sigma uncertainty for fossil fuel emissions is 16% of the fossil fuel emissions and 38% of the
biomass burning emissions in each month and each model grid box. The uncertainty of the ocean
flux is assumed to be 44%, and for both gross primary production and total ecosystem respiration
we assumed an uncertainty of 22% in each 3 hour time step and in each model grid.

ObsPack NRT was used, but observations from 'SCT', 'STR', 'TPD', 'PUY', 'KAS', and 'SSL'
were removed.

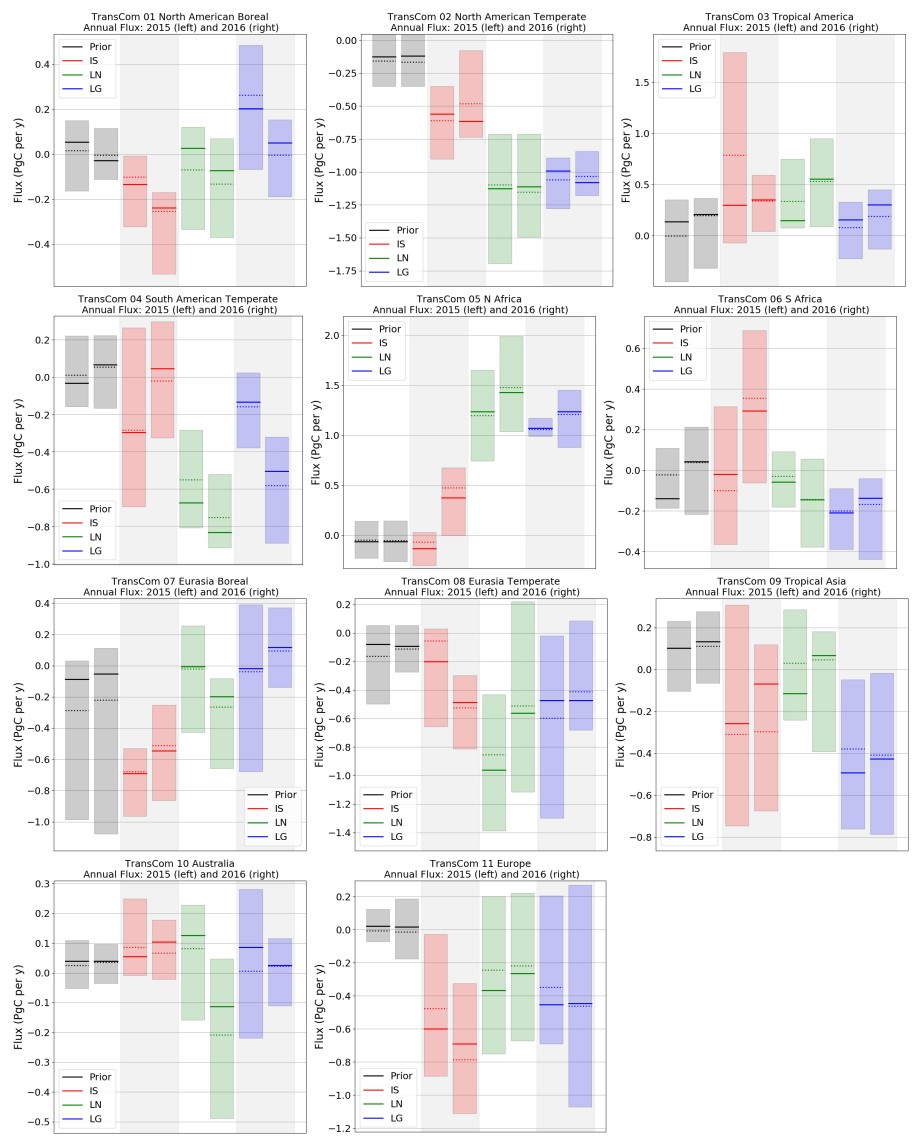

**Figure B1.** Ensemble annual fluxes for the 11 Transcom land regions. The bars each represent the trimmed range of the model ensemble posterior fluxes for 2015 (left bar) and 2016 (right bar). The solid line represents the median, and the dotted line represents the mean. The colors denote the prior fluxes (grey), as well as the posterior fluxes constrained by *in situ* (IS, red), land nadir (LN, green), and land glint (LG, blue) data.

## Appendix B: Level 4 Transcom Region Fluxes

Figures B1-B4 depict both annual and monthly fluxes for Transcom regions (Gurney et al., 2002). These are provided for direct comparison to previous literature, and so that the reader can easily seek out specific regions of interest.

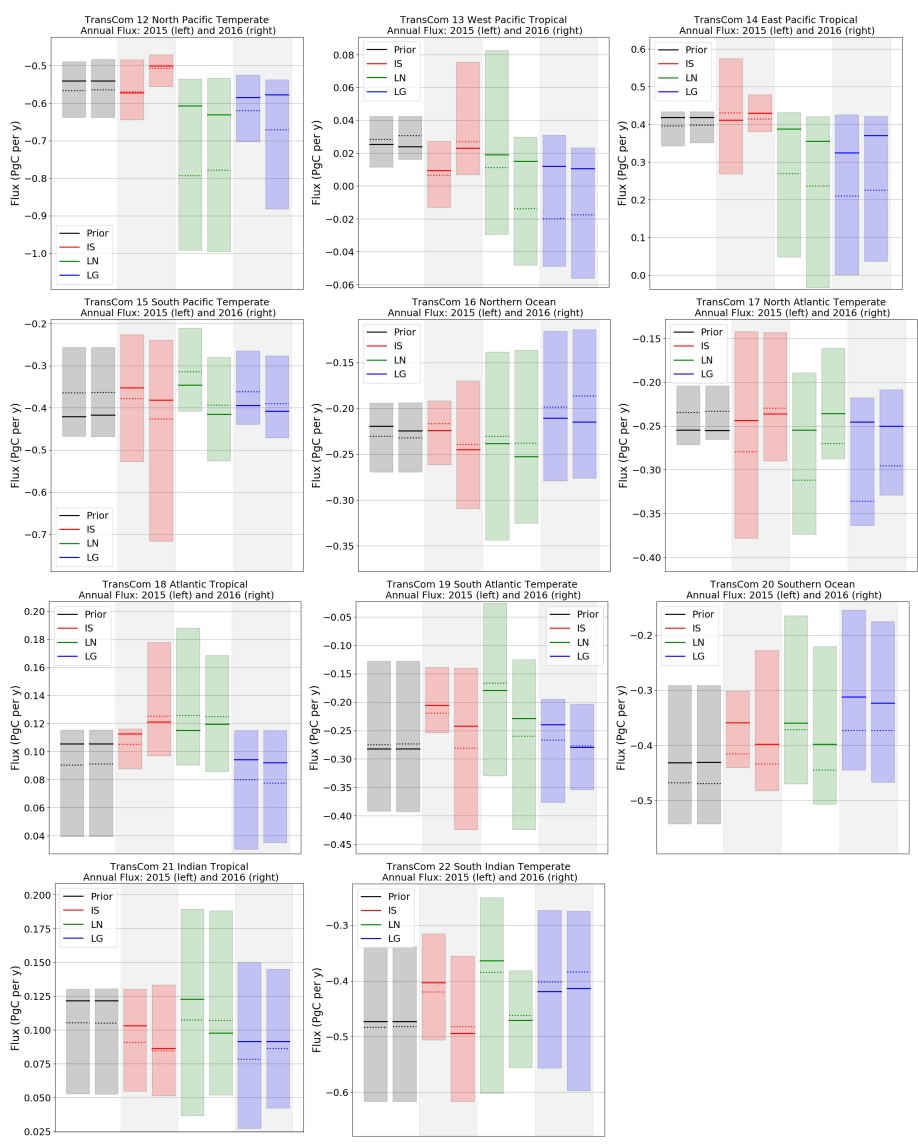

**Figure B2.** As in Figure B1, but for the 11 Transcom ocean regions.

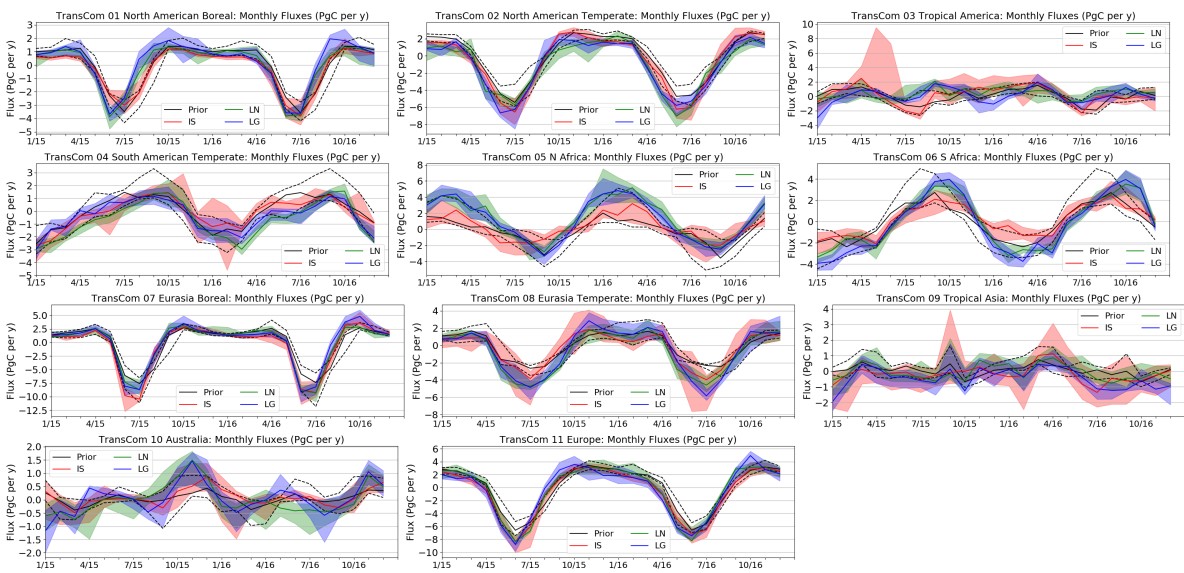

**Figure B3.** As in Figure B1, except that the fluxes are by month.

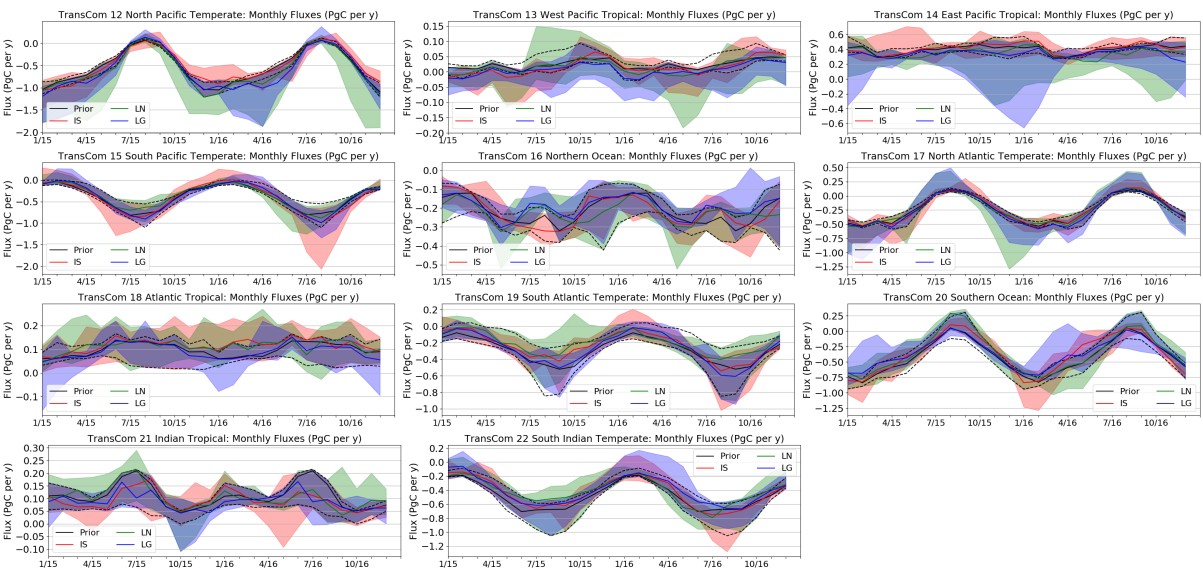

**Figure B4.** As in Figure B3, but for the 11 Transcom ocean regions.

**Appendix C:  Comparisons against TCCON**

TCCON $X_{CO_2}$ was binned to 30 minute averages as follows:

1. We first filtered all retrievals by TCCON's own quality flag to select only "good quality" retrievals, and to classify them by site and date.

2. For each day at each site, we fit a function of the form $\alpha\cos(\omega t+\phi)+\beta$ through the remaining retrievals, where $t$ is the local solar time (LST) in hours, $\omega = 2\pi/(24\text{hours})$, and $\alpha$, $\beta$ and $\phi$ are free parameters to be fit.

3. We calculate $\sigma$, the standard deviation of the residuals from the fit, and reject the sounding with the largest residual if it is more than $3\sigma$ away from the fit function. Then we recalculate the function fit with the updated set of retrievals, and repeat until no more retrievals are being rejected by the $3\sigma$ cutoff.

4. If at any stage the number of remaining soundings in a day falls below 3, or the total time spanned by the remaining soundings falls below 1 hour, we reject all soundings for that day.

5. If $\sigma >1$ppm for the remaining soundings, we reject all soundings for that day.

6. Once this outlier selection is done, we reject soundings with solar zenith angle $SZA > 60°$, and average the remaining soundings in 30 minute windows. The window edges are aligned to integer and half hours of the LST. The SZA is likewise averaged, and then used to look up the averaging kernel according to the TCCON prescription.

Our outlier filtering and averaging helps us create a dataset which is more appropriate for comparing to coarse resolution global models, which are unlikely to reproduce local $X_{CO_2}$ fronts and high frequency features. Figure C1 shows our filtering and averaging in action on a typical day's TCCON retrievals at Park Falls.

Comparisons of posterior simulated concentrations to TCCON data are given in this section as time series of residuals. An example of the TCCON data used for comparison from a single day at Park Falls, Wisconsin is shown in Figure C1. For ease of viewing, TCCON sites are split into three regions in Figures C2-C4.

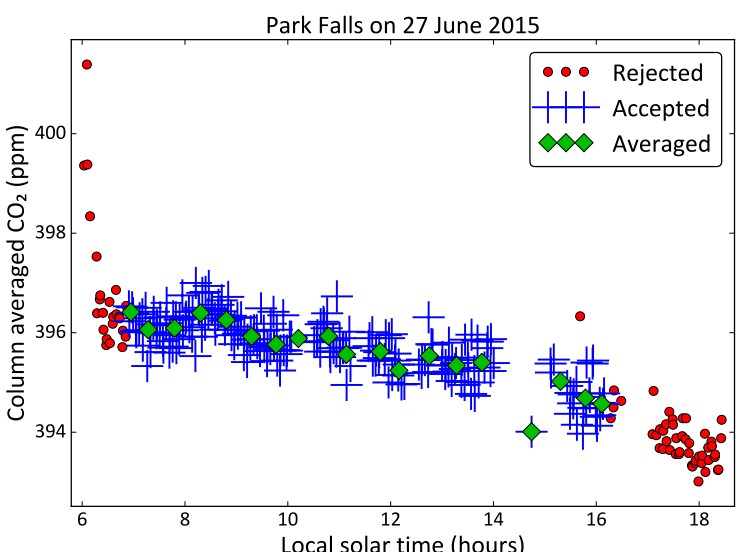

**Figure C1.** An example of TCCON $X_{CO_2}$ retrievals at Park Falls on June 27, 2015. Red circles denote retrievals that were rejected by the outlier filter, SZA filter and TCCON flagging, while blue plus signs denote retrievals that passed those filters. Green diamonds denote the 30 minute averages of the accepted retrievals that were eventually used by the modelers for this study.

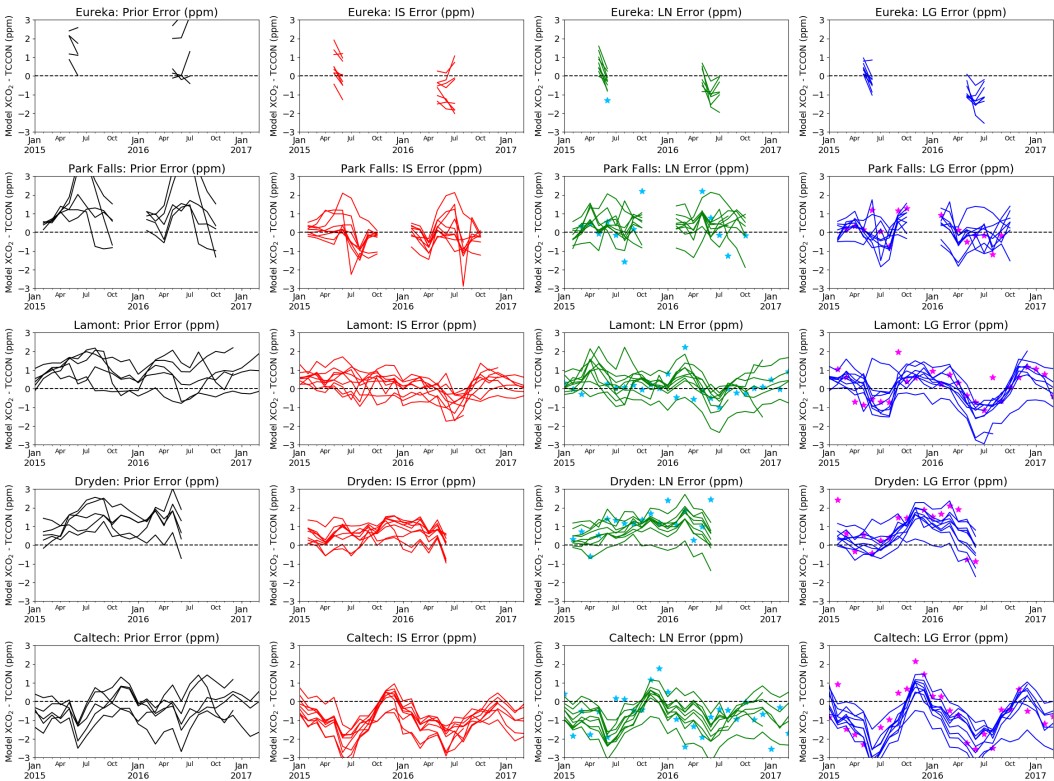

**Figure C2.** The time series of monthly mean residuals between simulated $X_{CO_2}$ and TCCON observed $X_{CO_2}$ by site and data constraint for sites in North America. Each line represents a different model. The sites are arranged from north to south by site latitude. The colors denote the prior concentrations (grey), as well as the posterior concentrations from forward runs using fluxes constrained by *in situ* (IS, red), land nadir (LN, green), and land glint (LG, blue) data. For the LN and LG residuals, monthly OCO-2 overpass residuals are displayed as stars over the model residuals. Plots are ordered by site latitude.

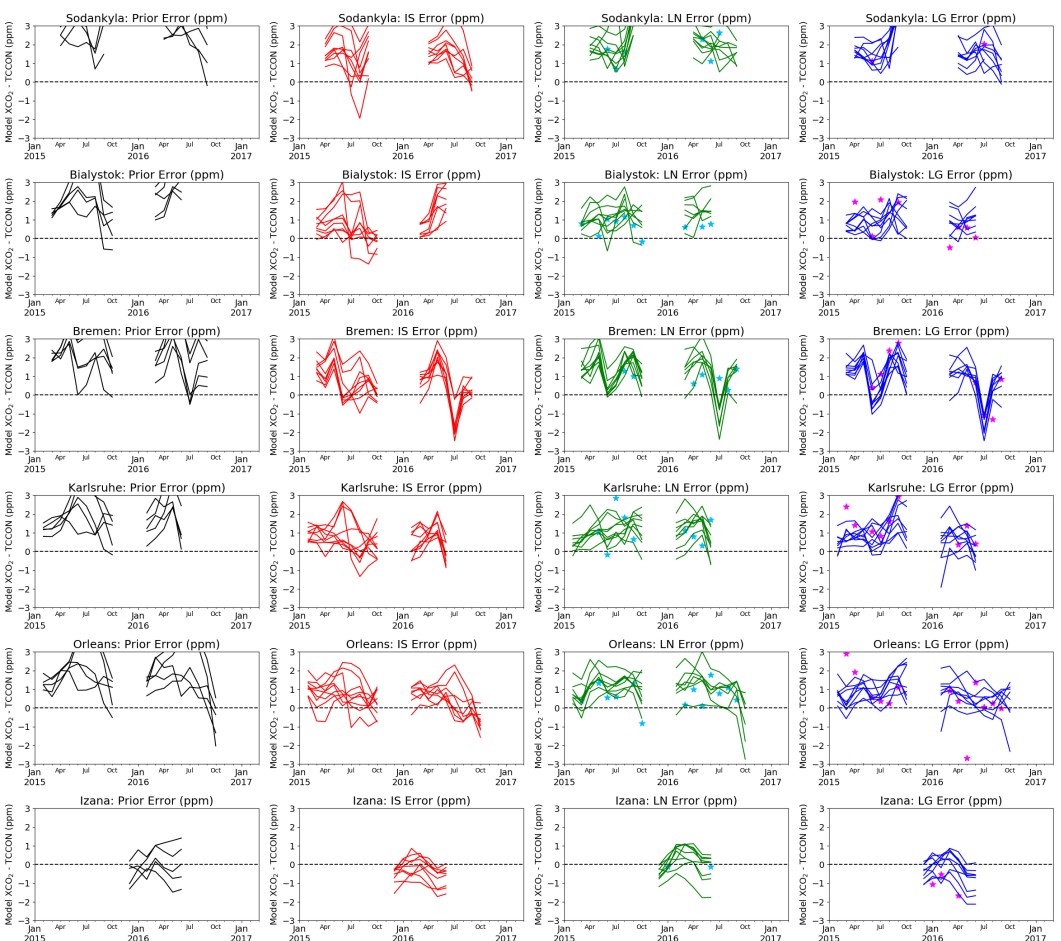

**Figure C3.** As in C2, but for European TCCON sites.

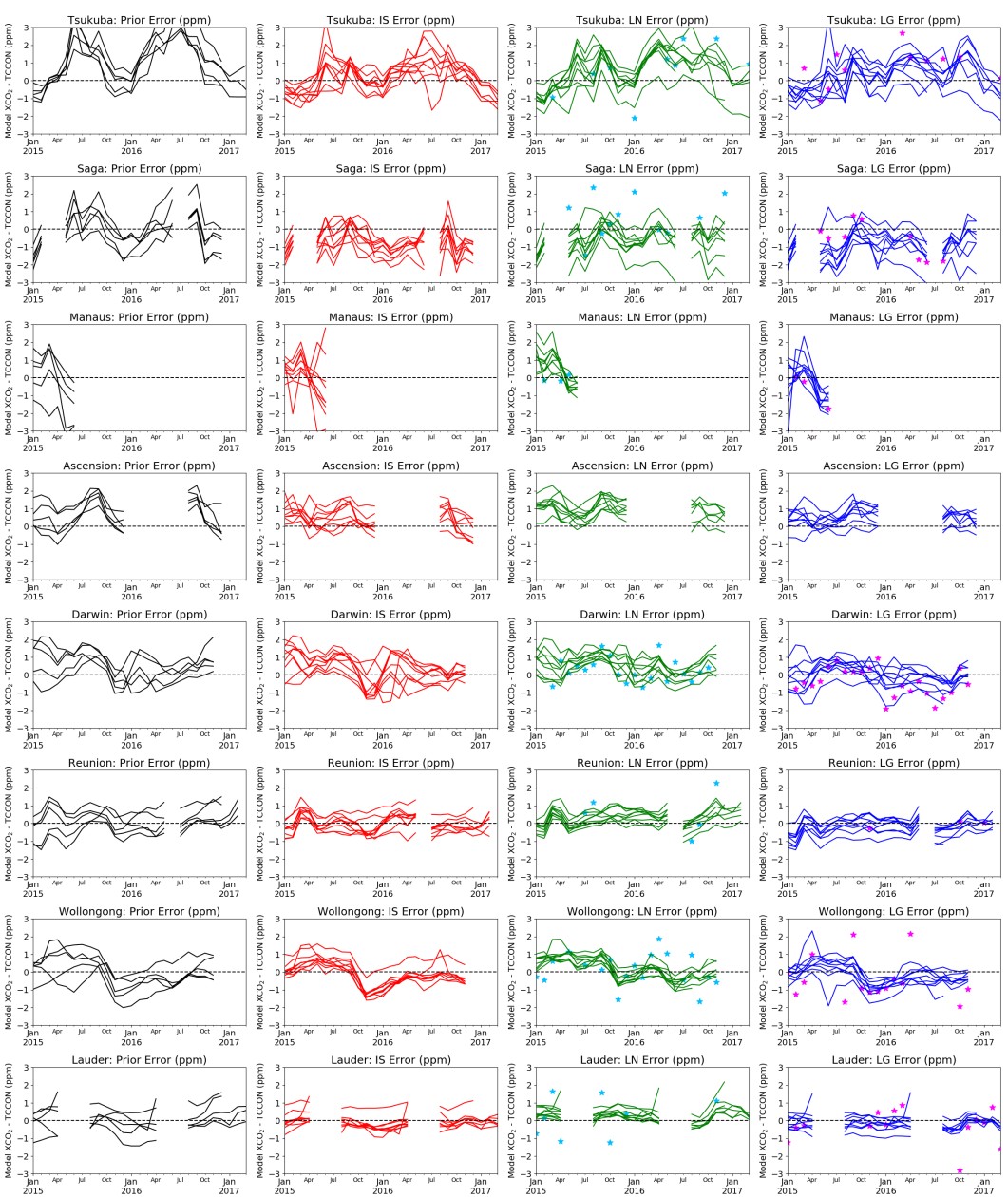

**Figure C4.** As in C2, but for sites in Japan, the tropics, and the southern hemisphere.