# Peer review of "The 2015-2016 Carbon Cycle As Seen from OCO-2 and the Global *In Situ* Network"

_Atmospheric Chemistry and Physics, 2019_

## Referee Comment (RC1) · Anonymous Referee #1 · 6 Mar 2019

This manuscript explores the usefulness of OCO-2 data using a multi-model data assimilation/inversion framework. The manuscript is generally well written, except for some missing information at places or the page formatting that makes reading of the Figures difficult. It appeared to me, at the very end (page 45), that the authors are using a very old version of OCO-2 retrieval. That's a major cause for concern. I do not know how that affect the results presented in this study and their interpretations. Also I do not find much new information from this work, beyond what we already know from the existing publications using remote sensing data. However, the authors have done a commendable job in bringing together several models and have taken heavy workload in analysing them. In view of the above, I recommend publication of the manuscript as "Technical note" only in ACP or transfer to "Atmospheric Measurement Techniques".

[Figure]

The scientific value of this paper is limited for publication a normal paper in ACP.

Specific comments: Page 3, lines 25ff: You may not be able to answer all these question, because the satellites do not see any parts of the globe seasonally uniformly (owing to the clouds in tropics and sunlight in the high latitudes. The data gap issue is not addressed here.

Page 4, line 20: I do not think this is only due to biases in the GOSAT, you may not simply discuss annual total CO2 flux from Sun following satellite sensor - as the winter seasonal data are missing.

Page 4, line 23ff: Give a reference (in prep. is ok) or this doesn't make sense. Delete?

Page 6, line 5-7: How can you say that? Is there a common transport model using different assimilation technique?

Page 6, line 17-21: I tend to agree that use of single model is problematic for the flux assessments, but any given model should be rigorously evaluated. For instance, I would highly recommend you to simulate SF6 or the likes as a dynamical tracer in all the models participating any such inter-comparisons, and present simulated horizontal and vertical gradients of SF6 in comparison with measurements. Note that when you are working with XCO2 the stratosphere is not negligible. Many studies have shown this clearly since 2012.

Page 8, line 17: what are land glint? how much fraction of land data are in glint mode. some additional information will be useful here. Please cite to Figure 1 here.

Page 11, line 8: How much of the para below is essential for this work?

Section 3.3: Should be shorten with some effort, these are pre-processing all global modellers do, but how much is essential is not clear. For example if you sampled the model for each of the data points and then took mean, will that work? If not, the whole paper raises an issue of applicability of the models for the purpose. Why not invest resources to develop models first and then estimate fluxes from satellite. For sure the

dilemma you face here is not new, but need serious attention.

Table and Figure : Please, fix the page formatting. Its annoying to find the figure/table and captions on different pages

Page 19, line 19: I do not think so, at least for the 2nd peak in 2016

Page 10, line 1: Need to define H15 and P13 somewhere?

Page 23, line 17-22: Strange discussion. Why not the data are flagged before inversion. How do you know that your model transport is not at fault?

Page 24: Somehow this shouldn't be the case when there are supposed to be data coverage everywhere! is this an illusion because of the variable axis range? Can you compare the uncertainties in number and be more precise in your discussion here?

Page 27: line 2: I think this and earlier statements like this are loosely supported by numbers. You are saying ∼1 PgC/yr values are similar in seasonal cycle, but a few fraction of a PgC/yr as different for annual flux!

Page 28: bottom row: TransCom 7 & 8? or 8 only?

Page 30, line 11: what if you do not measure during the wet season due to clouds?

Page 30, line 18-20: I do not think these smaller number are more difficult to interpret that the very large emission you mentioned earlier in this paragraph.

Figure 5: what are TransCom 03a, 05b or 09a? Show the region map in Fig. 1, you have wasted space for one more panel

Page 43: I thought one of the "Science" papers using OCO2 data has discussed this already? May be no need for a mention here.

Page 45: line 11: Aren't more and more people using LAI, rather than NDVI?

Page 45: line 23: I see this possibility as "remote" and out of place in the context of OCO-2 data inversion/assimilation. Suggest a deletion of APO related text.

A5: no mention of transport model resolution?

Figure C2: Why doesn't the OCO2 values match with the model even after assimilation? these two sentenses be merged - "The sites are arranged..." and "Plots are ordered..."

[Figure]

---

## Referee Comment (RC2) · Anonymous Referee #2 · 8 Mar 2019

Crowell et al. present results from an ensemble of atmospheric inversion modelling systems inverting column averaged XCO2 observations from OCO-2 over the years 2015-16. They find that on global scale the inferred fluxes are consistent with results from in situ CO2 inversions, however, on smaller regional scales, and especially in the tropics, the ensemble yields a large spread in the flux estimates.

The ensemble approach employed here is a major effort in understanding the short-comings in current inversion systems and identifying robust carbon sources and sinks. However, the main outcome of this research essentially is that the inverse modelling community is still not in the position of providing robust flux estimates at regional scales even with the advent of having a huge increase in the observations provided by satel-lites. Admittedly the results here are based on a rather outdated version of the OCO-2

[Figure]

XCO2 retrievals, but I doubt that the spread will be smaller when using the most recent XCO2 product because here it is mainly based on the ensemble, ie. different transport models, inversion methods and uncertainty correlation assumptions. This in itself is not a new finding. The paper fails at going the next step identifying in more detail the causes for the large differences in the flux estimates at regional scales. This is mainly due to the rather weak evaluation of the model results. A more rigorous evaluation of the modelling systems clearly demonstrating the skills of each system would certainly have helped to put the results in perspective.

The manuscript is mostly well written and structured but at some places it reads a bit unpleasant, e.g. having the figure captions not directly with the figures.

Some additional points: P 3 Ll 6-8: Here you argue that the uncertainty in recent estimates was due to the lack of observations outside North America and Europe. But even with the large increase of observations outside these regions by OCO-2 you still find a similar spread with your ensemble.

P 5 L 21 and P 23 Ll 13-19: What is the purpose of land glint observations? Why do you use these data if there are apparent shortcomings with them?

P 6 L 6: The impact of assimilation methodology is an interesting point but it is not systemtically investigated in Peylin et al., 2013.

P 7 L 3: Shouldn't it be Table 1 here? There is no table referenced before.

P 9 Ll 1-2: What are the 'lite' files and how are 'good' retrievals characterized?

P 9 L 3: Which variables are used for the IVE? I assume you don't use time, latitude, longitude etc.

P 20 Ll 1-2: What are P13 and H15?

P 30 L5 and Figs 4-6: Maybe add a Figure of these regions in the appendix, not everyone is familiar with the Transcom regions.
* * *
P 45 L 15: 'note' instead of 'not'

---

## Author Comment (AC1) · 14 May 2019

General Response:

We thank the reviewers for their comments, and have attached a draft here that we think addresses most of the comments. We naturally disagree with the assertion that this should be a technical note, as we feel that we have contributed a lot to the scientific discussion with our intercomparison of OCO-2 derived fluxes and in situ derived fluxes for the years of 2015 and 2016. There are a few places where further clarification from the reviewers is sought. They are welcome to contact us directly or through the editors for a more in depth discussion.

Reviewer 1:

[Figure]

This manuscript explores the usefulness of OCO-2 data using a multi-model data as-similation/inversion framework. The manuscript is generally well written, except for some missing information at places or the page formatting that makes reading of the Figures difficult. It appeared to me, at the very end (page 45), that the authors are using a very old version of OCO-2 retrieval. That's a major cause for concern. I do not know how that affect the results presented in this study and their interpretations. Also I do not find much new information from this work, beyond what we already know from the existing publications using remote sensing data. However, the authors have done a commendable job in bringing together several models and have taken heavy workload in analysing them. In view of the above, I recommend publication of the manuscript as "Technical note" only in ACP or transfer to "Atmospheric Measurement Techniques".

The scientific value of this paper is limited for publication a normal paper in ACP.

Response:

With regard to the version of the data set, we can only say that organizing the scope of work represented in this intercomparison effort has taken about two years from start to finish, and the inversions were already completed when Version 8 data was re-leased. The period from completion of the inversion experiments to submission of the manuscript was largely taken up by processing the evaluation data to try to falsify one of the sets of fluxes in the tropics, without success.

Finally, we note that a new round of experiments is beginning soon and will utilize the latest version of the OCO-2 retrievals. *We argue the paper has significant scientific value, as it is the first to our knowledge to discuss the flux constraint provided by in situ data and OCO-2 retrievals side by side for the 2015-2016 time period. There are several regions that evince significant differences depending on what data type is assimilated such as Tropical Africa, in which a potential 1 PgC source of carbon is either present or not depending on which data set one uses. We admit that the ability to falsify either hypothesis is at present beyond the ability of the evaluation data we*

*have available to us, but that does not prevent the work presented here from being significant. Were the reviewer to provide the examples of remote sensing data they reference, we would be glad to comment on them.*

*With regard to the version of the data set, we can only say that organizing the scope of work represented in this intercomparison effort has taken about two years from start to finish, and the inversions were already completed when Version 8 data was released. The period from completion of the inversion experiments to submission of the manuscript was largely taken up by processing the evaluation data to try to falsify one of the sets of fluxes in the tropics, without success.*

*Finally, we note that a new round of experiments is beginning soon and will utilize the latest version of the OCO-2 retrievals.*

*Specific comments: Page 3, lines 25ff: You may not be able to answer all these question, because the satellites do not see any parts of the globe seasonally uniformly (owing to the clouds in tropics and sunlight in the high latitudes. The data gap issue is not addressed here.*

*Response:* We address this issue in the later section that gives background on the OCO-2 retrievals. We don't feel that the introduction is the place to introduce all possible difficulties with the observations.

*Page 4, line 20: I do not think this is only due to biases in the GOSAT, you may not simply discuss annual total CO2 flux from Sun following satellite sensor - as the winter seasonal data are missing.*

*Response:* It is true. We have added a caveat that seasonally varying coverage might have an influence on these results as well.

*Page 4, line 23ff: Give a reference (in prep. is ok) or this doesn't make sense. Delete?*

*Response:* This reference has been added.

[Figure]

*Page 6, line 5-7: How can you say that? Is there a common transport model using different assimilation technique?*

*Response:* The paper did not focus on the assimilation techniques. We removed this text, as it was more of an intuitive statement than a proven result.

*Page 6, line 17-21: I tend to agree that use of single model is problematic for the flux assessments, but any given model should be rigorously evaluated. For instance, I would highly recommend you to simulate SF6 or the likes as a dynamical tracer in all the models participating any such inter-comparisons, and present simulated horizontal and vertical gradients of SF6 in comparison with measurements. Note that when you are working with XCO2 the stratosphere is not negligible. Many studies have shown this clearly since 2012.*

*Response:* We agree and understand, and note that a separate effort involving comparisons with SF6 are in progress with many of the same models used in the OCO-2 MIP activity. Those results will be reported at a later date, and the conclusions will be related to the results in this paper.

*Page 8, line 17: what are land glint? how much fraction of land data are in glint mode. some additional information will be useful here. Please cite to Figure 1 here.*

*Response:* We have included more information on the number of soundings in each mode at the end of the first paragraph Section 3.1.

*Page 11, line 8: How much of the para below is essential for this work?*

*Response:* The work involved in collecting the data at individual sites and in coordinating between the individual site PI's is significant. Additionally, these data are now publically available, and thus we prefer to include this discussion so that readers are made aware.

*Section 3.3: Should be shorten with some effort, these are pre-processing all global modellers do, but how much is essential is not clear. For example if you sampled the*

*model for each of the data points and then took mean, will that work? If not, the whole paper raises an issue of applicability of the models for the purpose. Why not invest resources to develop models first and then estimate fluxes from satellite. For sure the dilemma you face here is not new, but need serious attention.*

*Response:* Moved details to the Appendix.

*Table and Figure : Please, fix the page formatting. Its annoying to find the figure/table and captions on different pages*

*Response:* This is an artifact of the draft vs. latex template formatting (and occurred during typesetting by Copernicus). This will be resolved in the formatted manuscript.

*Page 19, line 19: I do not think so, at least for the 2nd peak in 2016*

*Response:* The mean of the prior has a minimum value of -2 PgC per month in both 2015 and 2016, while the posterior fluxes range from -2.75 PgC per month and lower in the mean. We aren't quite sure what this comment refers to.

*Page 10, line 1: Need to define H15 and P13 somewhere?*

*Response:* These references have been listed in complete form.

*Page 23, line 17-22: Strange discussion. Why not the data are flagged before inversion. How do you know that your model transport is not at fault?*

*Response:* We cannot falsify the transport hypothesis, but the common behavior across inversions (and very different behavior from LN) suggests that the "signal" is present in the data rather than in transport. The data are not "flagged" beyond the quality filter because there is no obvious cutoff in data quality for any of these variables, but rather a more general sense that errors grow with larger airmasses due to poorly understood scattering effects when the path length is long.

*Page 24: Somehow this shouldn't be the case when there are supposed to be data coverage everywhere! is this an illusion because of the variable axis range? Can you*

*compare the uncertainties in number and be more precise in your discussion here?*

*Response:* The numbers for the amplitudes have been included in the discussion, as have the uncertainties.

*Page 27: line 2: I think this and earlier statements like this are loosely supported by numbers. You are saying âĞă1 PgC/yr values are similar in seasonal cycle, but a few fraction of a PgC/yr as different for annual flux!*

*Response:* We have tightened up our presentation and made the statements less vague with more concrete values.

*Page 28: bottom row: TransCom 7  8? or 8 only?*

*Response:* Thanks for catching this. The name was incorrect and has been corrected. Additionally, a new figure has been added (Figure 2) that identifies the regions by number, and new text has been added that identifies the regions by number in the main body.

*Page 30, line 11: what if you do not measure during the wet season due to clouds?*

*Response:* There are a nontrivial number of 10s average observations in this region each month during the wet season, though of course there are less. This is clear from Figures 3-5 of Eldering et al, 2017.

*Page 30, line 18-20: I do not think these smaller number are more difficult to interpret that the very large emission you mentioned earlier in this paragraph.*

*Response:* Yes you're right. We have added some descriptive text.

*Figure 5: what are TransCom 03a, 05b or 09a? Show the region map in Fig. 1, you have wasted space for one more panel*

*Response:* We added a new Figure 2, which has the map in it.

*Page 43: I thought one of the "Science" papers using OCO2 data has discussed this*

[Figure]

*already? May be no need for a mention here.*

*Response:* From the comment and rereading the text on Page 43, we are not sure what "this" refers to.

*Page 45: line 11: Aren't more and more people using LAI, rather than NDVI?*

*Response:* Leaf Area Index is actually a product that is usually based on NDVI or fPAR. You need a model to move from something like fPAR to LAI, where there is an assumption on the distribution of the canopy. For example, MODIS's LAI product depends upon MODIS fPAR product and a land categorization. The bigger point we are trying to make here is that "grasslands", especially in areas frequently covered by clouds, are notoriously difficult to model from satellite data. The biggest single problem is that growth typically occurs during the wet season when the areas are covered by clouds and not visible by satellites. Furthermore, most vegetation indices tend to saturate over extremely dense and productive canopies, only being able to essentially see the 2D projection of "greenness" from above. On the other hand, prognostic models model the grassland growth as a function of precipitation, which is often available as a a product such as GPCP. So, even though the prognostic models are far from perfect, they tend to do better in these areas.

*Page 45: line 23: I see this possibility as "remote" and out of place in the context of OCO-2 data inversion/assimilation. Suggest a deletion of APO related text.*

*Response:* Removed.

*A5: no mention of transport model resolution?*

*Response:* Text added.

*Figure C2: Why doesn't the OCO2 values match with the model even after assimilation? these two sentenses be merged - "The sites are arranged..." and "Plots are ordered..."*

*Response:* The atmospheric inversions match data in a least squares sense globally, and so may not match individual overpasses of TCCON sites exactly. Nonetheless, the posterior concentrations do seem to match with seasonal differences relative to TCCON. The last sentence is removed.

*Reviewer 2*

*Crowell et al. present results from an ensemble of atmospheric inversion modelling systems inverting column averaged XCO2 observations from OCO-2 over the years 2015-16. They find that on global scale the inferred fluxes are consistent with results from in situ CO2 inversions, however, on smaller regional scales, and especially in the tropics, the ensemble yields a large spread in the flux estimates.*

*The ensemble approach employed here is a major effort in understanding the short-comings in current inversion systems and identifying robust carbon sources and sinks. However, the main outcome of this research essentially is that the inverse modelling community is still not in the position of providing robust flux estimates at regional scales even with the advent of having a huge increase in the observations provided by satellites. Admittedly the results here are based on a rather outdated version of the OCO-2 XCO2 retrievals, but I doubt that the spread will be smaller when using the most recent XCO2 product because here it is mainly based on the ensemble, ie. different transport models, inversion methods and uncertainty correlation assumptions. This in itself is not a new finding. The paper fails at going the next step identifying in more detail the causes for the large differences in the flux estimates at regional scales. This is mainly due to the rather weak evaluation of the model results. A more rigorous evaluation of the modelling systems clearly demonstrating the skills of each system would certainly have helped to put the results in perspective.*

*Response: We understand the perception that is presented with the small amount of evaluation data that was presented in the paper. In preparing this work, we evaluated posterior concentrations against a significant number of observations, including surface*

*data, aircraft from the NOAA network, numerous field campaigns, CONTRAIL, and others, and the results were actually difficult to distinguish by model across different assimilation constraints. The evaluation we show in the paper is a summary of what amounted to more than 6 months of attempting to falsify the various flux estimates. The evaluation data used includes the TCCON network, as well as all in situ data available to us at the surface and aboard aircraft. If the reviewer has suggestions of independent data, we welcome them.*

*It is worth mentioning here that a parallel effort on distinguishing between the models with SF6 data is currently in progress, and the results of that study will inform the conclusions of this manuscript.*

*The manuscript is mostly well written and structured but at some places it reads a bit unpleasant, e.g. having the figure captions not directly with the figures.*

*Response:* This is an unfortunate consequence of the typesetting after using the Copernicus template, which seems to be mismatched with the discussion draft. This will be fixed in the manuscript.

*Some additional points: P 3 Ll 6-8: Here you argue that the uncertainty in recent estimates was due to the lack of observations outside North America and Europe. But even with the large increase of observations outside these regions by OCO-2 you still find a similar spread with your ensemble.*

*Response:* This is a fair point. We have added some discussion if this point to the discussion section.

*P 5 L 21 and P 23 Ll 13-19: What is the purpose of land glint observations? Why do you use these data if there are apparent shortcomings with them?*

*Response:* The land glint observations are not obviously worse than the land nadir observations, and provide an alternate constraint on fluxes given their occurring at different times during the revisit cycle. The only apparent shortcomings show up at

high solar zenith angles, but analysis suggests that pinpointing the trigger for these effects is extremely difficult given the complexity of modeling aerosol scattering at long path lengths.

*P 6 L 6: The impact of assimilation methodology is an interesting point but it is not systemtically investigated in Peylin et al., 2013.*

*Response:* This is true, and our comment was based on a basic observation of two "clusters" of results that seemed to vary by assimilation methodology (i.e. synthesis vs. EnKF/4D-Var)

*P 7 L 3: Shouldn't it be Table 1 here? There is no table referenced before.*

*Response:* Correct. This has been fixed.

*P 9 Ll 1-2: What are the 'lite' files and how are 'good' retrievals characterized?*

*Response:* We have added some explanatory text here to clear it up. Much more detail is available in O'Dell et al (2018).

*P 9 L 3: Which variables are used for the IVE? I assume you don't use time, latitude, longitude etc.*

*Response:* The weighting is based on the reported L2 posterior uncertainties. We have modified the text for clarity.

*P 20 Ll 1-2: What are P13 and H15?*

*Response:* We corrected these to point to the full citation.

*P 30 L5 and Figs 4-6: Maybe add a Figure of these regions in the appendix, not everyone is familiar with the Transcom regions.*

*Response:* We added Figure 2 to clarify this.

*P 45 L 15: 'note' instead of 'not'*

*Response:* Corrected.

*Please also note the supplement to this comment: https://www.atmos-chem-phys-discuss.net/acp-2019-87/acp-2019-87-AC1- supplement.pdf*
* * *
*Interactive comment on Atmos. Chem. Phys. Discuss., https://doi.org/10.5194/acp-2019-87, 2019.*

[Figure]

**Supplement:**

Manuscript prepared for Atmos. Chem. Phys.

with version 2014/09/16 7.15 Copernicus papers of the LATEX class copernicus.cls. Date: 13 May 2019

**The 2015-2016 Carbon Cycle As Seen from OCO-2 and the Global *In Situ* Network**

Sean Crowell1, David Baker2, Andrew Schuh2, Sourish Basu3, Andrew R. Jacobson3, Frederic Chevallier5, Junjie Liu6, Feng Deng7, Liang Feng8,9 Kathryn McKain3, Abhishek Chatterjee10, John Miller4, Britton Stephens12, Annmarie Eldering6, David Crisp6, David Schimel6, Ray Nassar11, Christopher O'Dell2, Tomohiro Oda10, Colm Sweeney4, Paul I. Palmer8,9, and Dylan B. Jones7 1University of Oklahoma, Norman, OK, USA 2Cooperative Institute for Research in the Atmosphere, Colorado State University, Fort Collins, CO, USA 3Cooperative Institute for Research in Environmental Sciences, University of Colorado Boulder, Boulder, CO, USA 4NOAA Earth System Research Laboratory, Boulder, CO, USA 5Le Laboratoire des Sciences du Climat et de L'Environnement 6NASA Jet Propulsion Laboratory 7Department of Physics, University of Toronto 8School of GeoSciences, University of Edinburgh 9National Center for Earth Observation, University of Edinburgh

10Global Modeling and Assimilation Office, NASA Goddard Space Flight Center

11Climate Research Division, Environment and Climate Change Canada

12National Center for Atmospheric Research

Sprangendence to: Sean Crowell (scrowell@ou.edu)

[revised manuscript text omitted]